# Uncovering nitroxoline activity spectrum, mode of action and resistance across Gram-negative bacteria

Elisabetta Cacace [1,2], Manuela Tietgen[1], Meike Steinhauer[1], André Mateus[2], Tilman G. Schultze[1], Marina Eckermann[3,4], Marco Galardini [5,6], Vallo Varik [2], Alexandra Koumoutsi[2], Jordan J. Parzeller[1], Federico Corona[2], Askarbek Orakov [7], Michael Knopp[2], Amber Brauer-Nikonow [7], Peer Bork[7], Celia V. Romao [8], Michael Zimmermann [6], Peter Cloetens[3], Mikhail M. Savitski [2], Athanasios Typas [2] & Stephan Göttig [1] ✉

Nitroxoline is a bacteriostatic quinoline antibiotic, known to form complexes with metals. Its clinical indications are limited to uncomplicated urinary tract infections, with a susceptibility breakpoint only available for *Escherichia coli*. Here, we test > 1000 clinical isolates and demonstrate a much broader activity spectrum and species-specific bactericidal activity, including Gram-negative bacteria for which therapeutic options are limited due to multidrug resistance. By combining genetic and proteomic approaches with direct measurement of intracellular metals, we show that nitroxoline acts as a metallophore, inducing copper and zinc intoxication in bacterial cells. The compound displays additional effects on bacterial physiology, including alteration of outer membrane integrity, which underpins nitroxoline's synergies with large-scaffold antibiotics and resensitization of colistin-resistant *Enterobacteriaceae* in vitro and in vivo. Furthermore, we identify conserved resistance mechanisms across bacterial species, often leading to nitroxoline efflux.

The quinoline-derivative nitroxoline (8-hydroxy-5-nitroquinoline) (Fig. 1a) is an antibiotic, used for more than 50 years as treatment and prophylaxis of acute and recurrent urinary tract infections (UTIs) in several European and Asian countries[1–3]. Because of its excellent safety profile[4,6] and activity against different organisms, nitroxoline has been recently proposed to be repurposed as antituberculosis[7], antifungal[8,9], antiviral[10,11], antiparasitic[12] and anticancer[13–16] agent.

Nitroxoline has been shown to exert bacteriostatic activity against a broad range of Gram-negative and Gram-positive species[17,18].

However, a comprehensive characterisation of its activity spectrum across species and strains, including growth inhibition and killing, is missing. Despite its long-standing use, information on nitroxoline PK/PD profile remains sparse[19], with the only clinical breakpoint defined for *E. coli* from uncomplicated UTIs[2,20]. Nitroxoline can be associated with metal cations[21], thereby inhibiting biofilm formation[22,23] and metallo-β-lactamases[24,25]. However, it remains unclear (i) whether nitroxoline inhibits bacterial growth only by acting as a metal chelator, sequestering essential metals, or as a metallophore, leading to

[1]Goethe University Frankfurt, University Hospital, Institute for Medical Microbiology and Infection Control, Frankfurt, Germany. [2]European Molecular Biology Laboratory, Genome Biology Unit, Heidelberg, Germany. [3]European Synchrotron Radiation Facility (ESRF), Grenoble, France. [4]Institute of Applied Physics, University of Bern, Bern, Switzerland. [5]Institute for Molecular Bacteriology, TWINCORE Centre for Experimental and Clinical Infection Research, a joint venture between the Hannover Medical School (MHH) and the Helmholtz Centre for Infection Research (HZI), Hannover, Germany. [6]Cluster of Excellence RESIST (EXC 2155), Hannover Medical School (MHH), Hannover, Germany. [7]European Molecular Biology Laboratory, Structural and Computational Biology Unit, Heidelberg, Germany. [8]Instituto de Tecnologia Química e Biológica António Xavier, Universidade Nova de Lisboa, Oeiras, Portugal. ✉e-mail: goettig@med.uni-frankfurt.de

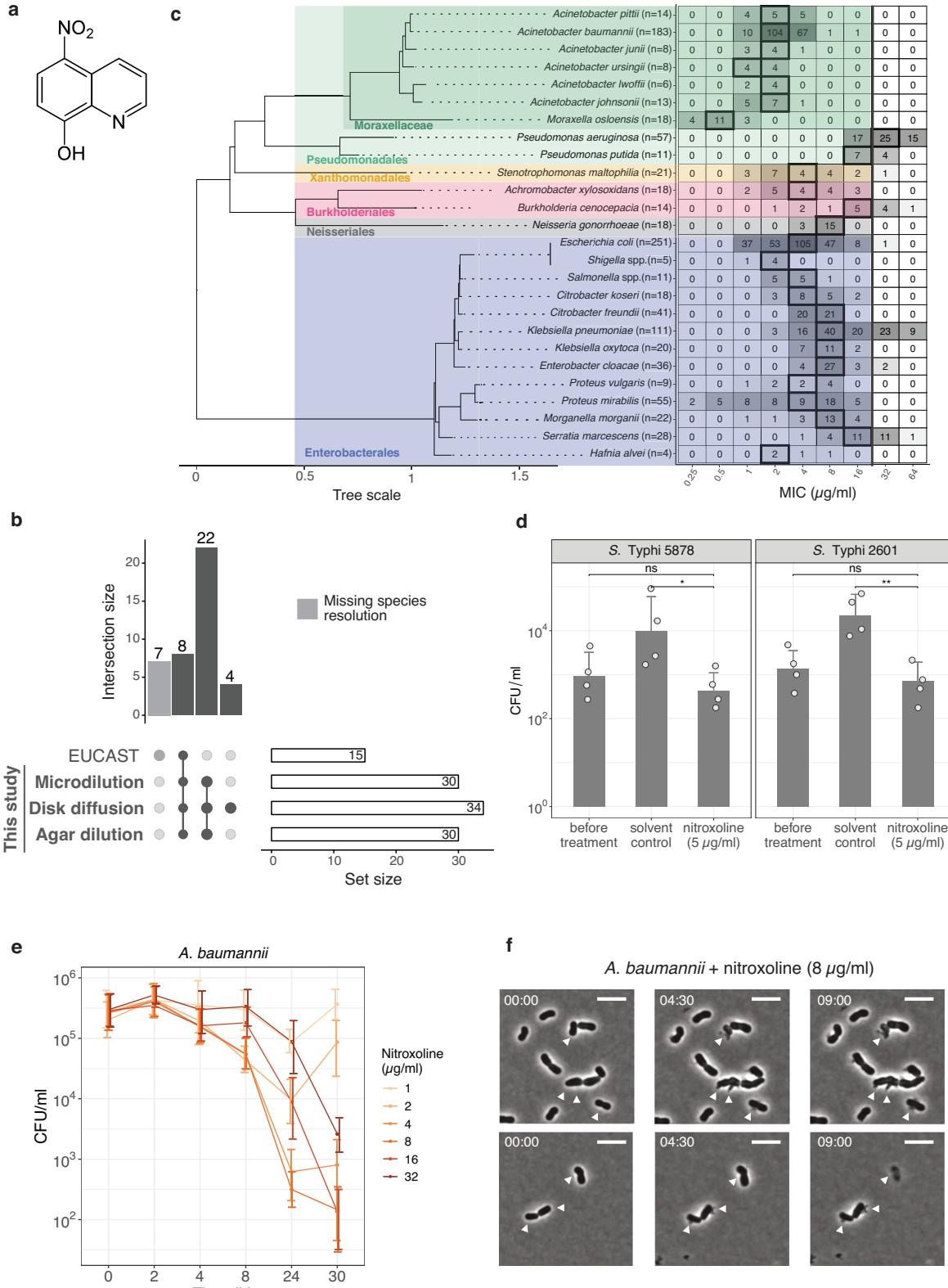

metal stress; (ii) which metals are differentially affected and (iii) whether it has other cellular effects dependent or independent of its perturbation of metal homoeostasis. Other quinoline derivatives have been proposed to affect the activity of RNA-polymerase by chelating $Mg^{2+}$ and $Mn^{2+}$ in yeasts[26] and in vitro on the isolated enzyme[27], and the activity of methionine aminopeptidases[4]. However, direct evidence of nitroxoline's effects on metal concentrations and a

systematic characterisation of its direct and indirect cellular effects is lacking.

Nitroxoline resistance seems to be rare in *E. coli*[28] patient isolates, with only a few resistance mechanisms identified to date in vitro. These include the overexpression of the tripartite efflux pump EmrAB-TolC as a result of first-step mutations in the *emrR* gene (encoding a transcriptional repressor of the pump), and

**Fig. 1 | Nitroxoline is active beyond UTI pathogens, including intracellular bacteria, and exerts bactericidal activity. a** Nitroxoline structure. **b** Overlap between Gram-negative bacterial genera or species tested with three orthogonal susceptibility testing methods in this study and according to EUCAST[18]. All seven EUCAST-specific entries are genera for which species resolution is missing. The number of overlapping species/genera between EUCAST and our methods is shown as intersection size, whereas the number of genera/species assessed by each approach is shown as set size. **c** Nitroxoline is active against several Gram-negative bacterial species. MICs were determined against 30 bacterial species in broth microdilution. The total number of strains tested is indicated next to species, ordered by phylogeny according to GTDB[77] (Methods). The clinical breakpoint for *E. coli* (16 μg/ml) is indicated (black line). $MIC_{50}$ values are framed in black and listed in the Source Data Fig. 1c together with $MIC_{90}$ values. **d** Nitroxoline is active against

intracellular *S*. Typhi. Intracellular bacterial counts were assessed with the gentamicin protection assay in two *S*. Typhi clinical isolates (Methods, Supplementary Data 2). Cell counts were determined before and after treatment with nitroxoline (5 μg/ml) or solvent control (DMSO) at 7 h p.i. (MOI 100). Mean and standard error are shown across four independent experiments. ns $p > 0.05$; *$p = 0.022$; **$p = 0.003$ (two-sided Welch's *t*-test). **e** Nitroxoline is bactericidal against *A. baumannii* ATCC 19606[T]. Mean and standard deviation across at least three biological replicates are shown for each condition. **f** Nitroxoline induces lysis in *A. baumannii* ATCC 19606[T]. White arrowheads mark the release of cytoplasmic material and loss of the pericellular halo. Representative images of phase-contrast videos were acquired after 8 μg/ml nitroxoline treatment (4x MIC, Methods, Supplementary Movie 1). The scale bar denotes 5 μm. Source data are provided as a Source Data file.

second-step mutations in *marR* and *lon*, conferring a higher *tolC* expression and resistance[29]. Mutations in *emrR* have also been identified in *K. pneumoniae*, as well as in the oqxRAB system[30], suggesting increased efflux as a key mode of nitroxoline resistance.

From these results, we propose nitroxoline as a broader antibiotic against many Gram-negative species. We challenge previous knowledge of nitroxoline as a bacteriostatic agent, demonstrating bactericidal activity on pathogens for which therapeutic options are limited, such as *A. baumannii*. We uncover strong synergies of nitroxoline with several antibiotics, including colistin, resensitizing colistin-resistant Enterobacteriaceae in vitro and in vivo. Combining systems-based approaches with direct measurements of outer membrane (OM) integrity and intracellular metal concentrations, we establish that nitroxoline acts as a zinc and copper metallophore and perturbs OM integrity. Finally, we show that the most recurring resistance mechanism across different pathogenic Enterobacteriaceae is the upregulation of Resistance-Nodulation-Division (RND) efflux pumps. Overall, we provide in vitro and in vivo evidence of nitroxoline's ability to act, alone or in combination, against hard-to-treat bacterial pathogens, and provide further mechanistic understanding of its mode of action and resistance.

## Results

### Nitroxoline has a broad activity spectrum against Gram-negative bacteria

Although nitroxoline (Fig. 1a) has been used for decades against uncomplicated UTIs and has a good safety profile[1,4–6], minimum inhibitory concentration (MIC) distributions are available from EUCAST only for eight bacterial species and seven genera[18], with a clinical breakpoint determined only for *E. coli*[2] (Fig. 1b). To investigate whether nitroxoline could be repurposed against other bacterial pathogens, we systematically profiled its susceptibility against 1000–1815 strains from 34 Gram-negative species. This set included some of the most relevant pathogens in the current antimicrobial resistance crisis, such as *A. baumannii*, *Burkholderia cenocepacia* complex and *Stenotrophomonas maltophilia* (Fig. 1b, c and Supplementary Fig. 1a, b).

We measured MICs in three orthogonal ways (Fig. 1b, Methods): broth microdilution (1000 isolates, Fig. 1c), disk diffusion (1815 isolates, Supplementary Fig. 1a) and agar dilution (1004 isolates, Supplementary Fig. 1b). We observed good concordance between this study and EUCAST[18] for the eight overlapping species (Supplementary Fig. 1c) and across the three methods, with the best agreement between broth and agar dilutions (Pearson correlation, $R = 0.94$) (Supplementary Fig. 1d–f).

From these results, we confirmed that several Gram-negative species were comparably or more susceptible to nitroxoline than *E. coli* ($MIC_{50} = 4$ μg/ml for broth and agar dilution, lower than the EUCAST breakpoint of 16 μg/ml for *E. coli*[2]). For *E. coli* we confirmed this value, except for one strain out of 251 clinical isolates with MIC twofold the breakpoint. Nitroxoline was active against Enterobacterales (median broth MIC: 4 μg/ml) and Moraxellaceae, which

had the lowest MIC values (median broth MIC: 2 μg/ml) and comprise pathogens for which therapeutic options are currently limited, such as *A. baumannii*. Only *P. aeruginosa* exhibited a $MIC_{50}$ of 32 μg/ml above the clinical breakpoint (Fig. 1c). Overall, these results indicate nitroxoline as a promising antibacterial option against Enterobacterales and *Acinetobacter* spp.

### Nitroxoline is active against intracellular *Salmonella* Typhi and bactericidal against *A. baumannii*

*Salmonella* spp. were among the most susceptible species to nitroxoline (average MIC in broth: 3.45 μg/ml; maximum MIC in broth: 8 μg/ml in only 1/11 tested strains) (Fig. 1c and Supplementary Fig. 1a, b). Since *Salmonella* can invade and persist in the host intracellularly[31], we tested whether nitroxoline could also act on intracellular *Salmonella* Typhi, the leading serovar responsible for enteric fever[32]. Using an in vitro infection assay (gentamicin protection assay[33], Methods), we showed that nitroxoline treatment results in a strong decrease (>97% compared to solvent control) of intracellular *S*. Typhi in infected HeLa cells, for both clinical isolates tested (Fig. 1d).

As other metal chelators[34], nitroxoline is classically considered bacteriostatic[35]. While we confirmed this in *E. coli* for concentrations eight times its average MIC in broth (32 μg/ml) (Supplementary Fig. 2a), we detected partial bactericidal activity (decrease of at least 3 $log_{10}$ colony-forming units (CFU) /ml at 24 h[36]) against *A. baumannii*, for concentrations as low as 8 μg/ml, i.e. three times its average MIC of *A. baumannii* in broth (2.79 μg/ml) (Fig. 1e and Supplementary Fig. 2b, Methods). Cells released their cytoplasmic content and lysed as early as 4.5 h of incubation with the drug (Fig. 1f and Supplementary Movies 1, 2, Methods). To our knowledge, this is the first evidence of nitroxoline's bactericidal activity.

### Nitroxoline antagonises beta-lactams and synergises with colistin

To explore potential combinatorial regimens, we tested nitroxoline in combination with 32 antibiotics in *E. coli* BW25113. The drug panel included all main classes of antibiotics used against Gram-negative bacteria, but also other metal chelators and antibiotics only effective against Gram-positive bacteria, such as vancomycin (Methods, Supplementary Data 1).

We uncovered extensive antagonism with beta-lactams (bactericidal cell-wall targeting drugs), including penicillins, cephalosporins and carbapenems (Fig. 2a and Supplementary Figs. 3, 4). Like other antagonisms between bactericidal and bacteriostatic drugs (such as nitroxoline at the concentration tested here), these interactions could be based on the fact that bactericidal drugs are more effective on actively dividing cells, and slowing down division with a bacteriostatic agent can alleviate their action[37].

One of the most potent synergies of nitroxoline was with the OM-targeting drug colistin (Fig. 2a and Supplementary Figs. 3, 4), whose toxicity limits its therapeutic use as a last-resort agent[38]. Nitroxoline could, therefore, be used to lower colistin concentrations

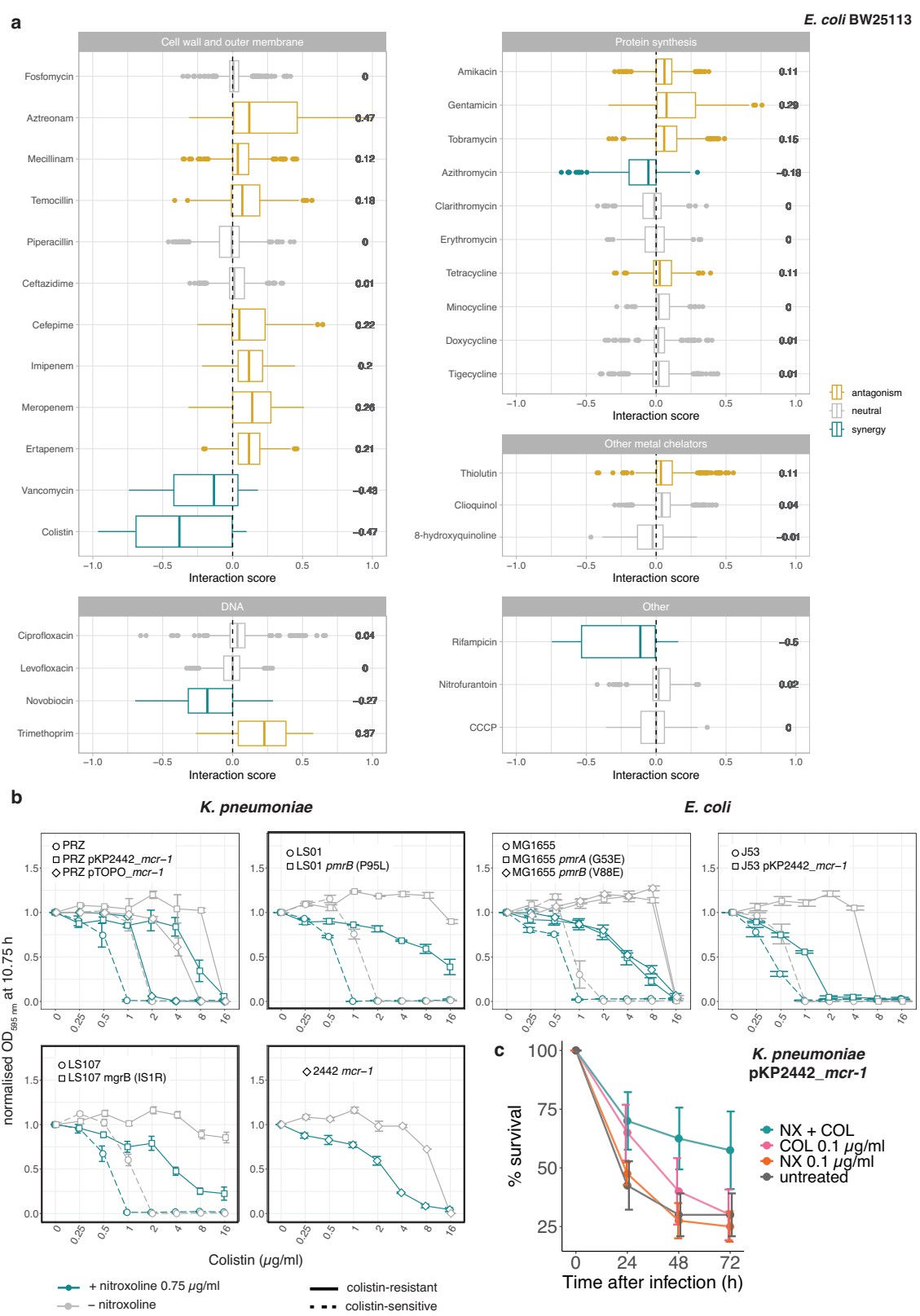

required to achieve therapeutic success, preventing toxicity. To explore this possibility, we tested whether nitroxoline could not only potentiate colistin action on sensitive strains, but also resensitize colistin-resistant strains. We showed that the addition of nitroxoline at sub-MIC concentration (0.75 µg/ml) decreases the MIC of *E. coli* and *K. pneumoniae* colistin-resistant strains (clinical and experimentally generated) from two- to four-fold, even below

colistin EUCAST breakpoint (2 µg/ml)[2] in three cases (Fig. 2b). To confirm this synergy in vivo, we infected *Galleria mellonella* larvae with an *mcr-1* positive, colistin-resistant *K. pneumoniae* clinical isolate (Methods, Supplementary Data. 2). The addition of nitroxoline improved the survival of the infected larvae by twofold at 72 h post-infection compared to the monotherapy with colistin or nitroxoline alone (Fig. 2c).

**Fig. 2 | Nitroxoline interacts with other antibiotics in *E. coli* and resensitizes colistin-resistant *E. coli* and *K. pneumoniae*. a** Nitroxoline interacts with several antibiotics in *E. coli*. Nitroxoline combinations were tested in 8 × 8 broth micro-dilution checkerboards in *E. coli* BW25113 (Supplementary Fig. 3). Bliss interaction score distributions are shown for each combination (*n* = 98 scores corresponding to 7 × 7 dose-combinations in two biological replicates). The median (central line), first (lower hinge) and third quartile (upper hinge) are shown for each boxplot. Whiskers correspond to 1.5x IQR from each hinge. The numbers stand for cumulative Bliss scores for each combination (Methods). **b** Nitroxoline resensitizes colistin-resistant *K. pneumoniae* and *E. coli*. Growth (OD$_{595\,nm}$ at 10.75 h, corresponding to the beginning of stationary phase for the untreated control for each strain) was measured in the presence of serial twofold dilutions of colistin, supplemented or not with 0.75 µg/ml nitroxoline and normalised by no-drug controls.

Three *K. pneumoniae* and two *E. coli* strains (dashed lines) and their isogenic colistin-resistant descendants (solid lines) were tested, including experimentally evolved and clinical isolates (framed in black, Supplementary Data 2). One *K. pneumoniae* clinical isolate carries the *mcr-1* positive natural plasmid pKP2442 and, therefore, lacks a parental strain. Mean and standard error across four biological replicates are shown. **c** Nitroxoline resensitizes a colistin-resistant *K. pneumoniae* clinical isolate in vivo. *G. mellonella* larvae were infected with the indicated isolate and treated with single drugs, their combination or were left untreated (solvent-only control). The mean and standard error are shown across four independent experiments for each condition. *p* = 0.0255 and *p* = 0.0098 comparing colistin-nitroxoline with colistin and untreated, respectively (log-rank test). NX nitroxoline, COL colistin. Source data are provided as a Source Data file.

## Nitroxoline perturbs the OM in *E. coli*

In addition to colistin, nitroxoline synergised with all large-scaffold antibiotics, including drugs that are normally excluded by the OM and therefore are not active against Gram-negative bacteria, such as macrolides, rifampicin, novobiocin and vancomycin (Fig. 2a and Supplementary Fig. 3). Altogether, this suggested a direct effect of nitroxoline on the OM permeability. To obtain a broader view of nitroxoline's direct and indirect effects, we performed two-dimensional thermal proteome profiling (2D-TPP)[39] on *E. coli* BW25113 to monitor the abundance and stability of proteins upon nitroxoline exposure. TPP is based on the principle that changes in the interactions of a protein (e.g. with a drug) can lead to changes in its thermal stability[40]. We exposed bacteria (whole-cell) or lysates to multiple nitroxoline concentrations and subjected them to a gradient of temperatures, capturing nitroxoline effects on both protein abundance and stability (Methods). While changes in lysates will typically only detect direct target(s) of drugs, as the biochemical environment of the cell has been disrupted, whole-cell changes provide a snapshot of both direct and indirect effects. We could not detect any significant change in lysate samples, suggesting that nitroxoline does not directly target a protein. In whole-cell samples, we observed a decrease in the abundance/stability of OM proteins (OMPs) and members of LPS biosynthesis and trafficking (Lpt) machinery (Fig. 3a, Supplementary Fig. 5a, b and Supplementary Data 3). These effects are similar to those caused by genetic perturbations known to influence OM stability[41].

We combined this data with chemical genetics, in which we systematically mapped nitroxoline effects on the fitness of deletion mutants of every non-essential gene in *E. coli*[42,43]. We found that mutants involved in similar processes, including LPS biosynthesis and transport, were more sensitive to nitroxoline, except for mutants involved in the first reactions of heptose incorporation into LPS inner core biosynthesis (*gmhA*, *gmhB* and *waaC*), which were more resistant (Fig. 3b, Supplementary Fig. 5c and Supplementary Data 4). This suggests the heptosyl-Kdo$_2$ moiety of the lipopolysaccharide inner core as a minimum requirement for nitroxoline activity, since deletion of mutants catalysing most downstream LPS biosynthetic reactions, starting with *waaF*, are more sensitive to nitroxoline.

To provide direct evidence of nitroxoline's effect on the OM, we quantified OM disruption using the hydrophobic probe 1-*N*-phenylnaphthylamine (NPN), which emits fluorescence upon exposure of the OM phospholipid layer[42] (Methods). Sub-MIC concentrations of nitroxoline resulted in a significantly higher fluorescence than control samples (unexposed to any drug or to the non-OM-affecting antibiotic chloramphenicol). As positive controls, we used OM-targeting antibiotic polymyxin B and EDTA, another metal chelator that disrupts OM by sequestering LPS-stabilising divalent cations[43]. Although lower than in positive controls, OM disruption by nitroxoline occurred even at 1/9 MIC (0.225 µg/ml, MIC = 2 µg/ml in *E. coli* BW25113, Fig. 3c).

To corroborate nitroxoline's action on the OM, we tested its activity against the OM-defective *E. coli* strain *lptD4213*[44] and in OM-perturbing conditions (0.5% SDS and 0.8 or 0.4 mM EDTA)[45] using an

efficiency of plating (EOP) assay (Methods). The *lptD4213* mutant was more susceptible to nitroxoline than wild-type *E. coli* (Supplementary Fig. 5d), in agreement with LptD decreased stability (Fig. 3a and Supplementary Fig. 5b) and with its loss-of-fitness already observed in the chemical genetic data, where the *lptD4213* mutant was included[42] (Fig. 3b). Furthermore, nitroxoline synergised with the OM-perturbing conditions at concentrations tenfold lower than MIC (Fig. 3d and Supplementary Fig. 5e).

It is possible that nitroxoline acts similarly to EDTA, chelating metals necessary for the stability of the OM[46]. While EDTA action is based on chelation of both Mg$^{2+}$ and Ca$^{2+}$, the two main cations involved in LPS stability[47], nitroxoline has been shown to preferentially complex with Mn$^{2+}$ and Mg$^{2+}$, with variable reports on the effect of Ca$^{2+}$ supplementation on MIC[21,48]. This might explain nitroxoline's smaller effect than EDTA on OM integrity (Fig. 3c). However, the abundance of OMPs and Lpt machinery proteins was also altered (Fig. 3a and Supplementary Fig. 5a, b), suggesting an effect of nitroxoline on the regulation of the levels of these proteins.

## Nitroxoline acts as a zinc and copper metallophore

Nitroxoline is reported to chelate Mn$^{2+}$ and Mg$^{2+}$ and reach the intracellular milieu[21], but a broader and more resolved view of its effects on metal homoeostasis is missing. From the 2D-TPP (Fig. 3a and Supplementary Fig. 5a) and chemical genetic data (Fig. 3b and Supplementary Fig. 5c), we identified distinct profiles for proteins involved in the import, intracellular utilisation, and export of metals, consistent with responses to copper (Fig. 4a) and zinc (Fig. 4b) increase.

At nitroxoline concentration around MIC (2 µg/ml), we observed an increased abundance of the copper-exporting P-type ATPase CopA (twenty-fold) and the multicopper oxidase CueO (fivefold), which upon Cu(I) excess remove copper from the cytoplasm and oxidise it in the periplasm, respectively[49] (Fig. 4a and Supplementary Data 3). Accordingly, the deletion mutant of *cueR*, which encodes the positive regulator of *copA*[50], is more sensitive to nitroxoline (Fig. 3b). A similar effect has been reported for other quinolines, forming complexes with copper[51,52], which may be extruded in the periplasm by CopA as a copper-overload defence mechanism. We observed similar effects for zinc, with the stabilisation of the zinc-responsive regulators Zur and ZntR and consistent changes in Zur- and ZntR-regulated proteins, such as subunits of the zinc importer ZnuABC (repressed by Zur) and the exporter ZntA (positively regulated by ZntR)[53] (Fig. 4b). Accordingly, ΔzntR was more sensitive to nitroxoline, whereas we could not detect significant fitness changes in *znuABC* deletion mutants (except for a slight decrease for ΔznuB). This suggests that the limitation of zinc uptake is not sufficient to confer nitroxoline resistance, and that other mechanisms (e.g. zinc efflux, active drug efflux) have a stronger impact on fitness upon nitroxoline exposure (Fig. 3b and Supplementary Data 4).

Zinc and copper intoxication has been associated with the disruption of iron-sulfur (FeS) clusters[54–57] and a compensatory induction of iron-uptake proteins (derepressed by Fur), and FeS cluster

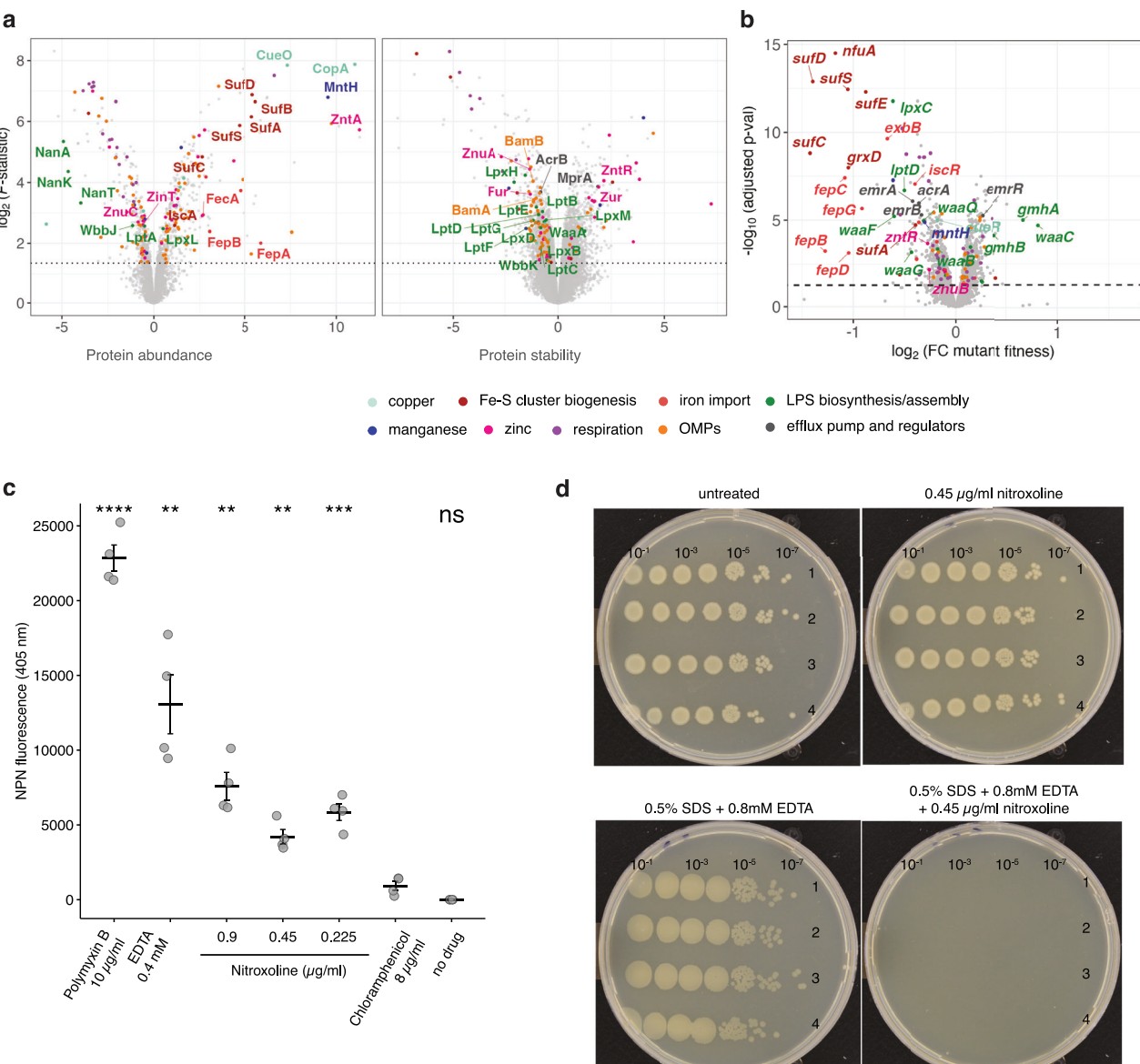

**Fig. 3 | Nitroxoline directly perturbs the OM in *E. coli*. a** Nitroxoline decreases the abundance and stability of outer membrane proteins and Lpt machinery. Volcano plots depicting abundance (left) or stability (right) changes upon nitroxoline exposure in whole-cell 2D-TPP. Results are based on *n* = 5 independent experiments (four drug concentrations and a vehicle control). Effect size and statistical significance as log₂ (*F*-statistic) (Methods) are represented on the x- and y-axis, respectively. The F-statistic was transformed to 1 when 0 before the log₂ transformation. Proteins are colour-coded according to their Gene Ontology (GO) annotation (Supplementary Fig. 5a). **b** Nitroxoline effects profiled by chemical genetics on an *E. coli* whole-genome single-gene deletion mutant library[43]. Effects are expressed as multiplicative changes of mutant fitness compared to the plate median (approximating wild-type). Significance was obtained from an empirical Bayes' moderated two-sided *t*-statistics, Benjamini–Hochberg adjusted (two independent clones per mutant, three replicates per condition, Methods,

Supplementary Data 4). Genes are colour-coded as in Fig. 3a (GO in Supplementary Fig. 5c). **c** Nitroxoline directly affects OM permeability. NPN fluorescence upon exposure of *E. coli* BW25113 to nitroxoline, positive (polymyxin B, EDTA) and negative (chloramphenicol, untreated samples) controls. Data points represent the average for each of the four biological replicates per condition. The horizontal line and error bars indicate mean and standard error. ns *p* > 0.05; \*\**p* ≤ 0.01; \*\*\**p* ≤ 0.001; \*\*\*\**p* ≤ 0.0001 (two-sided Welch's *t*-test using the chloramphenicol control as the reference group). EDTA, *p* = 0.008; Polymyxin B, *p* = 0.00004; no-drug control, *p* = 0.053; nitroxoline 0.45 µg/ml, *p* = 0.0008; nitroxoline 0.9 µg/ml, *p* = 0.003; nitroxoline 1.8 µg/ml, *p* = 0.002. **d** Nitroxoline is more potent upon chemical perturbation of the OM. EOP assays with tenfold serial dilutions of *E. coli* BW25113 cells plated onto no-drug control plates, 0.5% SDS + 0.8 mM EDTA, 0.45 µg/ml nitroxoline, or their combination. Four biological replicates were tested for each condition. Source data are provided as a Source Data file.

biogenesis (*suf* genes, induced by IscR). Accordingly, upon nitroxoline exposure, we observed decreased stability of Fur and an associated increase of Fur-repressed proteins involved in enterobactin biosynthesis (EntABF), recycling (Fes), receptor (FepA) and importing system (FepBCDG)[58]. The stability of IscR increased, with the associated decrease of *isc* operon and increase of *suf* operon members[59] (Fig. 3a and Supplementary Fig. 5f). The increased abundance of the manganese importer MntH (Figs. 3a, 4c) has also been reported as a

consequence of copper stress[60], which is consistent with the increased sensitivity of the corresponding mutant (Fig. 3b). Since MntH is repressed by Fur, its increase is consistent with the observed Fur destabilisation (Supplementary Fig. 5f).

To confirm the impact of these effects on intracellular metal concentrations, we performed synchrotron-based nano-X-ray-fluorescence (XRF) on nitroxoline-treated and untreated *E. coli*, confirming a four-fold copper, twofold zinc, and tenfold manganese

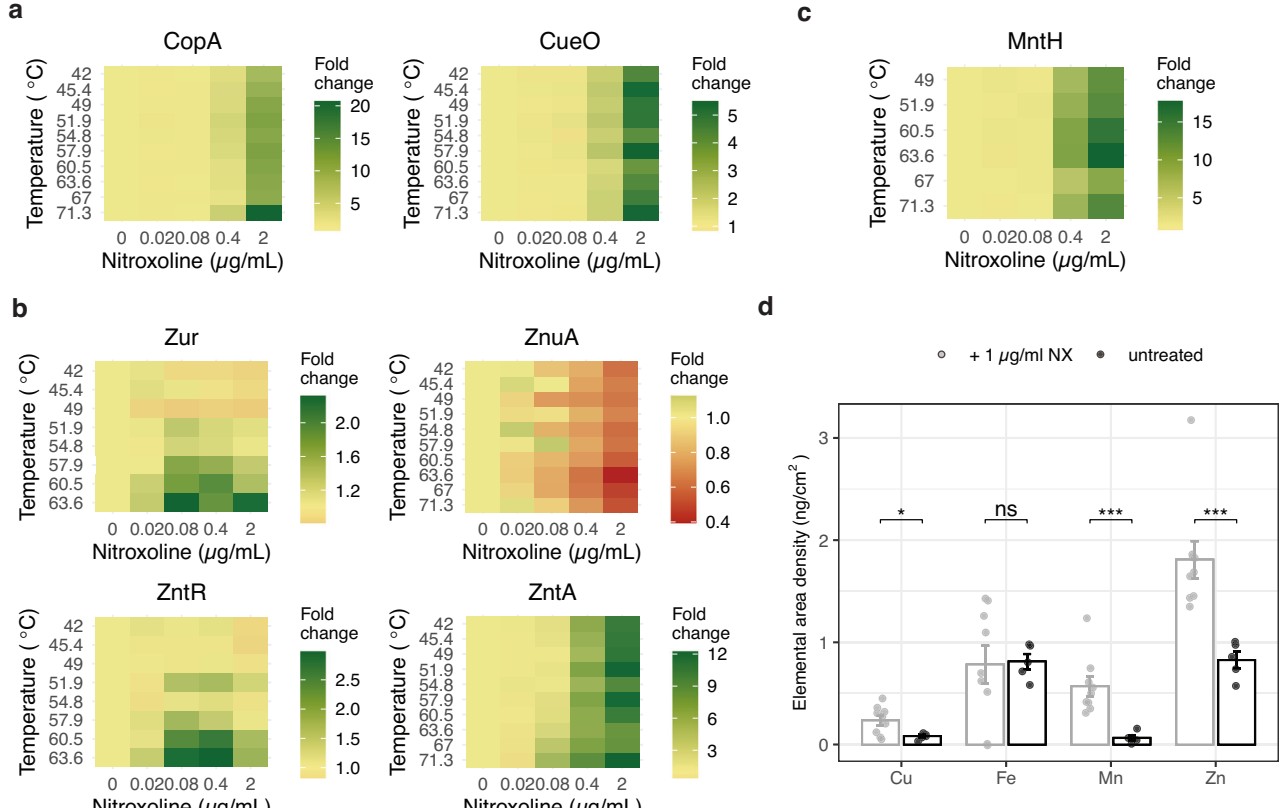

**Fig. 4 | Nitroxoline increases intracellular levels of copper and zinc.**
**a–c** Nitroxoline affects metal homoeostasis inducing copper and zinc detoxification responses, as determined by 2D-TPP. Heatmaps show the relative remaining soluble fraction compared to the vehicle control at each temperature to highlight changes in protein abundance and thermal stability profiles of the Cu(I) exporter CopA, the periplasmic copper oxidase CueO (**a**), the transcriptional regulators Zur and ZntR, zinc importer ZnuA and exporter ZntA (**b**), and the manganese importer

MntH (**c**), are shown. **d** Nitroxoline increases intracellular levels of copper, zinc and manganese. Synchrotron-based nano-XRF measurements on *E. coli* untreated or exposed to nitroxoline (1 μg/ml), expressed as elemental areal density (ng/cm²). The mean and standard error across ≥5 cells are shown (Methods). ns $p > 0.05$; *$p \leq 0.05$; **$p \leq 0.01$; ***$p \leq 0.001$ (two-sided Welch's *t*-test). Copper, $p = 0.01$; iron, $p = 0.73$; manganese, $p = 0.0006$; zinc, $p = 0.0005$. Source data are provided as a Source Data file.

increase in treated cells (Fig. 4d). Overall, our data suggest pleiotropic effects of nitroxoline on metal homoeostasis, consistent with its activity as ionophore for copper, previously reported for clioquinol in cancer cells[52], and zinc, as shown for other quinolines[61].

## Nitroxoline resistance is based on conserved mechanisms across species

Our results suggest that nitroxoline does not have a direct protein target, but rather exerts pleiotropic effects on OM integrity and metal homoeostasis, which might underpin the previously observed low frequency of resistance[1,6,22,28,30]. To explore resistance mechanisms across different species, we evolved resistance to nitroxoline in vitro in three species: *E. coli*, for which nitroxoline is already used, and two Gram-negative species, *K. pneumoniae* and *A. baumannii*, for which nitroxoline could be repurposed considering its low MIC (Fig. 1c and Supplementary Fig. 1a, b). We calculated the frequency of resistance in two strains for each species. Overall, across the three species, we found 19/24 and 21/24 lineages with frequency of resistance below 10⁻¹⁰ at four times and eight times the MIC, respectively, confirming previous reports on *E. coli* and *K. pneumoniae*[1,6,22,28,30] and showing this for the first time for *A. baumannii* (Supplementary Data 5). This further supports the repurposing of nitroxoline on a broader range of Gram-negative bacteria.

We performed whole-genome sequencing (WGS) on 12 *E. coli*, 8 *K. pneumoniae* and 6 *A. baumannii* sensitive and evolved resistant strains (fold increase MIC ≥4 compared to parental-sensitive strain) (Methods, Fig. 5a and Supplementary Data 2), and performed proteomics on a

subset of them (Fig. 5b, Supplementary Fig. 6a, b and Supplementary Data 6). Mutations across species primarily affected transcriptional repressors of RND-type efflux pumps: *emrR* (previously reported in *E. coli*[29]), *oqxR* (*K. pneumoniae*), *adeL* and *tetR/acrR* (*A. baumannii*) (Fig. 5a).

We found *emrR* mutations in all 12 evolved nitroxoline-resistant *E. coli* strains (Fig. 5a). This is consistent with previous reports[29] and our chemical genetic data, where the *emrR* deletion mutant was more resistant, and the knockouts of its regulated pump *emrAB* were more sensitive to nitroxoline (Fig. 3b). To verify the clinical relevance of these mutations, we assessed them in 14 clinical isolates with reduced susceptibility (MIC ≥8 μg/ml, i.e. at least two times the MIC₅₀ measured in this study for *E. coli*), finding distinct mutations from experimentally evolved strains (Supplementary Fig. 6c). We also found *emrR* mutations in two *K. pneumoniae* nitroxoline-resistant strains, which, as previously shown for *E. coli*[29], had higher EmrA and TolC protein levels. An *A. baumannii* resistant strain, although lacking any mutation of efflux pump regulators, also exhibited a fourfold increase in EmrA levels (Fig. 5b and Supplementary Data 6).

Surprisingly, we did not detect any increase in EmrA and only a slight (<2-fold) increase of TolC in three *E. coli* *emrR*-mutated strains, which showed instead a decreased abundance of porins OmpD, OmpF and LamB (Fig. 5b). For at least two of these strains (2_R1 and 2_R4), this could depend on missense mutations of *envZ*, that regulates porin expression via OmpR[62,63], increased in these strains (Fig. 5b). Alternatively, porin abundance changes could be explained by mutations in the *lon* gene (Fig. 5a), previously associated with nitroxoline resistance[29] and resulting in the stabilisation of the Lon protease

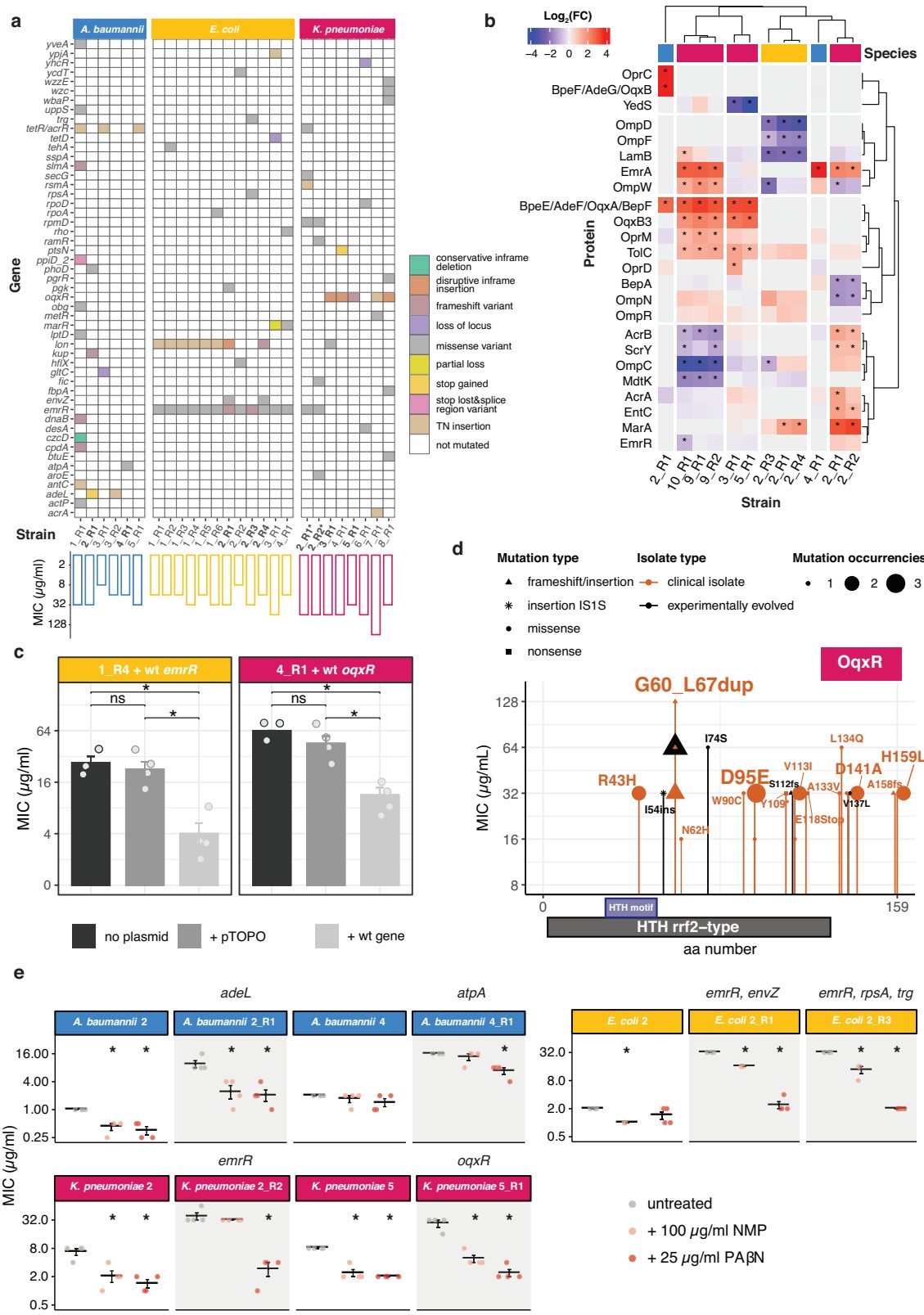

substrate MarA[64]. which regulates the expression of several drug resistance determinants, including porins[65,66], and is also increased in these strains (Fig. 5b). Given the unexpected proteome changes in *emrR*-mutated *E. coli*, we sought to confirm the functional relevance of these mutations, complementing a strain carrying a recurring missense mutation (D109V, Supplementary Fig. 6c) with wild-type *emrR*, thereby restoring nitroxoline susceptibility (Fig. 5c).

The most frequent genetic alterations in *K. pneumoniae*-resistant strains were mutations in *oqxR*, the transcriptional repressor of the OqxAB efflux pump, in agreement with recent reports[30]. We identified *oqxR* mutations in 5/8 experimentally evolved *K. pneumoniae* strains and in all 14 clinical isolates sequenced (Fig. 5d). We identified a mutational hotspot, common to clinical isolates and experimentally evolved strains: a duplication of eight amino acids (G60_67dup)

**Fig. 5 | Resistance to nitroxoline is associated with efflux pump upregulation across species. a** Whole-genome sequencing of experimentally evolved nitroxoline-resistant strains (Supplementary Data 2). Nitroxoline MIC values are indicated below each strain. Mutation effects are colour-coded. Strains on which proteomics was performed (Fig. 5b) are indicated in bold. *K. pneumoniae* strains whose sensitive parental strain lacks *oqxR* are marked with an asterisk. The in-patient evolved *K. pneumoniae* clinical isolate 8_R1 is indicated in italics. **b** Protein abundance changes in nitroxoline-resistant strains. Selected proteins annotated as efflux pumps or porins are shown and clustered according to Pearson's correlation. Hits are marked with an asterisk (adjusted *p* value ≤0.05 and at least twofold abundance change (Supplementary Fig. 6a, b and Supplementary Data 6). Species are colour-coded, as in Fig. 5a. **c** Wild-type *emrR* and *oqxr* complementation restores nitroxoline susceptibility. The experimentally evolved *E. coli* strain 1_R4 with *emrR* D109V mutation (Fig. 5a and Supplementary Fig. 6c) and *K. pneumoniae* strain 4_R1 harbouring the *oqxR* G60-L67 duplication (Fig. 5a, d and Supplementary

Fig. 6d) are shown. Nitroxoline MIC was measured by broth microdilution. Mean and standard error across four biological replicates are shown. ns $p > 0.05$; *$p \leq 0.05$ (Wilcoxon test. For *E. coli*: empty plasmid vs no-plasmid, $p = 0.608$; empty plasmid vs wild-type efflux pump, $p = 0.042$; no-plasmid vs wild-type efflux pump, $p = 0.042$. For *K. pneumoniae*: empty plasmid vs no-plasmid, $p = 0.217$; empty plasmid vs wild-type efflux pump, $p = 0.042$; no-plasmid vs wild-type efflux pump, $p = 0.042$). **d** Amino acid changes resulting from *oqxR* mutations. The domain annotation of OqxR was obtained from its closest annotated structural homologue NsrR (Methods). **e** Efflux pump inhibitors resensitize nitroxoline-resistant strains. Nitroxoline MIC was measured by broth microdilution. Mean and standard error across four biological replicates are shown. Resistant strains are shown as shaded plots next to their parental-sensitive strains. For results on all strains, see Supplementary Fig. 6g. *p* values are only shown when significant: *$p \leq 0.05$; **$p \leq 0.01$ (Wilcoxon test). Source data are provided as a Source Data file.

resulting in a loop addition (Fig. 5d and Supplementary Fig. 6d). This mutation also emerged in a patient after four-month prophylaxis with nitroxoline (*K. pneumoniae* urine isolate 8_R1, Fig. 5a), confirming its relevance for in vivo evolution of resistance. Accordingly, *oqxR*-mutated strains coclustered in the proteomics data (Fig. 5b and Supplementary Fig. 6b) and showed an increased abundance of OqxA (BepF), OqxB (OqxB3) and TolC (Fig. 5b and Supplementary Data 6). To further demonstrate the impact of this duplication on resistance, we complemented a G60_67dup-positive strain with wild-type *oqxR*, restoring nitroxoline susceptibility (Fig. 5c).

In resistant *A. baumannii* the most common mutations affected two transcriptional regulators: *adeL*, repressing the expression of the efflux pump AdeFG(BepF)-OprC, and a transcriptional regulator of the *acrR/tetR* family (Fig. 5a and Supplementary Fig. 6e, f). We performed proteomics on a strain carrying an *adeL* mutation resulting in a premature stop codon, which accordingly showed an increased abundance of all components of the efflux pump AdeFG-OprC (Fig. 5b). Another resistant strain, although not carrying any mutation in efflux pump regulators, exhibited a fourfold increase in EmrA abundance, which could explain its resistance (Fig. 5b).

From the mutational spectrum and proteomic changes that we observed across species, increased drug efflux via RND pumps appeared as a conserved strategy to achieve nitroxoline resistance. To verify this hypothesis, we tested the impact of the efflux pump inhibitors (EPI) 1-(1-naphthylmethyl)-piperazine (NMP) and phenylalanine-arginine β-naphthylamide (PAβN) on nitroxoline susceptibility. We observed a decrease in nitroxoline MIC both in resistant and susceptible strains, independent of their specific mutations and generally more marked for PAβN, previously reported as an inhibitor of RND pumps in *E. coli*[67] and of AdeFG in *A. baumanni*[68] (Fig. 5e and Supplementary Fig. 6g, Methods).

## Discussion

The alarming spread of antimicrobial resistance is aggravated by the slow development of new compounds. This is not only due to the experimental challenge of developing novel compounds, ideally with novel bacterial targets and low resistance potential, but also to economic hurdles in bringing new compounds to the clinic. In this context, repurposing already approved drugs, with known PK/PD and toxicity profile, holds great potential to accelerate the clinical translation of novel antibacterial strategies.

With nitroxoline, we show how revisiting the spectrum and mode of action of an FDA-approved drug opens new therapeutic possibilities for some of the most challenging bacterial species in the current AMR scenario like colistin-resistant Enterobacteriaceae and *A. baumannii*. In addition to its usage as single drug, we demonstrate nitroxoline as a powerful synergizer in combination with other drugs, sensitising *E. coli* to antibiotics normally bottlenecked by the OM and active only against Gram-positive species. Additionally, nitroxoline resensitized colistin-

resistant Enterobacteriaceae, independently of the species and the colistin resistance determinant, including in vivo against an *mcr-1* positive *K. pneumoniae* clinical isolate. While further studies are needed to verify nitroxoline's adequate therapeutic concentrations beyond its current UTI indications, our results, as well as recent anticancer formulations for other anatomical regions than the bladder[14,69,70], suggest that the activity that we demonstrate in vitro and in vivo could also be achieved in humans for novel therapeutic uses. We also uncovered generalised antagonism with beta-lactams, whose in vivo relevance should be validated in future studies and may have important implications for combination therapies.

Despite its decade-long use, the mode of action of nitroxoline has remained elusive. Here, we revisit its activity with systems-biology approaches, such as chemical genetics, 2D-TPP and high-throughput drug combinatorial testing. The integration of this data uncovered new effects advancing our understanding of nitroxoline's mode of action beyond metal chelation. Nitroxoline has been reported to preferentially associate with $Mn^{2+}$ and $Mg^{2+}$, with variable effects of $Ca^{2+}$ supplementation on the MIC[21,48]. This may underpin nitroxoline's OM damage (Fig. 3c), since the two main cations required for LPS stability are $Mg^{2+}$ and $Ca^{2+}$[47], similarly to other OM-disrupting cation chelators like EDTA. However, nitroxoline also altered the abundance and stability of OMPs and members of the Lpt machinery (Fig. 3a, b and Supplementary Fig. 5a–e). This suggests that nitroxoline may damage the OM not only via extracellular chelation of LPS-stabilising cations, but also by indirectly disrupting LPS biosynthesis and OMPs. By profiling nitroxoline's effects at genome and proteome levels in *E. coli*, we were able to uncover metal-specific effects, including a novel activity as copper and zinc metallophore, which we showed indirectly from defence mechanisms against these metals' toxicity (Fig. 3a, b), and directly via nano-X-ray fluorescence (Fig. 4d). This effect may be relevant in other Gram-negative species, as suggested by conserved responses to nitroxoline exposure, including the increase of the copper and zinc exporters, of siderophores (likely as a response to the damage of FeS clusters by metal stress), and of phenols and polyamines, known to act as antioxidants, particularly upon metal intoxication[71] (Figs. 3a, b, 4a–c and Supplementary Figs. 7, 8).

Considering our results, the broad antibacterial spectrum against Gram-negative bacteria could be attributed to the fact that nitroxoline seems to lack a specific protein target, with pleiotropic effects on the OM and on metal homoeostasis. Despite nitroxoline's broad spectrum, we identified critical species-specific differences, such as bactericidal activity in *A. baumannii*. This challenges the definition of nitroxoline as a bacteriostatic agent and points towards envelope damage. Although we could also demonstrate such damage in *E. coli* (Fig. 3c), we could not recapitulate similar microscopic changes (cell bursting, release of cytoplasmic content) with microscopy at nitroxoline concentrations close to MIC (Supplementary Movie 3). This may imply that these effects are dose-dependent and could also be observed in *E. coli* at

higher concentrations, where killing is observed (eight times MIC, 32 µg/ml) (Supplementary Fig. 2a). These differences could be due to differences in OM composition[72] and/or defence from metal stress[73] across the two species, largely unexplored. Importantly, since drugs can be bacteriostatic or bactericidal depending on the strain considered[74], future studies should explore the conservation of such activity across multiple *A. baumannii* strains.

We also revealed cross-species mechanisms for resistance, such as the regulation of efflux pumps, part of the RND superfamily prevalent in Gram-negative bacteria. A few shared responses were previously observed for another quinoline, chloroxine[51], including the increase of MarA, also resulting in efflux pump upregulation or porin downregulation (as we showed for *E. coli*) and of the nitroreductases NfsA and NfsB. While this could suggest cross-resistance between nitroxoline and nitrofuran anti-biotics, for which this is the most common resistance determinant, this has been previously disproven, at least in *E. coli*[29]. Importantly, neither through resistance evolution nor in naturally occurring resistant isolates could we detect mutations on potential protein targets, in concordance with previous reports[29] and with our 2D-TPP results, which did not identify any protein stabilisation in lysates. This supports the absence of a specific protein target for nitroxoline and excludes, to the best of our knowledge, an important potential resistance mode.

In summary, we show how revisiting a compound used for dec-ades with systems approaches can reveal a novel spectrum, mode of action and resistance mechanisms, offering new and safe therapeutic possibilities against hard-to-treat bacterial species.

## Methods

### Bacterial strains and growth conditions
All strains used in this study are listed in Supplementary Data 2. Unless otherwise specified, bacteria were grown in cation-adjusted Müller-Hinton (MH II) broth at 37 °C with continuous shaking at 180 rpm in 5 ml for overnight cultures and in 50 µl in microtiter plates. For growth on a solid medium, MH II was supplemented with 1.5% agar.

### MIC measurement and efflux pump inhibitor supplementation
Antimicrobial susceptibility was determined by disc diffusion (Kirby Bauer assay), agar dilution (nitroxoline concentration range: 0.125–128 µg/ml) and broth microdilution (range: 0.125–64 µg/ml) as previously described[75]. MICs and zones of inhibitions were evaluated and interpreted according to EUCAST[2]. Efflux pumps, 1-(1-Naphthyl-methyl)-piperazine (NMP) and phenylalanine-arginine β-naphthylamide dihydrochloride (PAβN) were diluted in 25 g/l stock solutions in DMSO and used as previously described[76].

### Species phylogeny analysis
A phylogeny tree was constructed from the Genome Taxonomy Database (GTDB) bacterial reference tree (release 08-RS214[77]) using the ETE toolkit[78]. The GTDB taxonomy decorating the tree was then used to convert genome IDs to their corresponding species names. The tree was visualised using the R package ggtree[79].

### Time-kill curves
A bacterial suspension in 0.9% NaCl (McFarland standard of 0.5) was prepared from overnight cultures, diluted 1:100 in 10 ml of MH II broth and incubated at 37 °C with continuous shaking for 30 h with a twofold dilution series of nitroxoline (1/2-16x MIC), or DMSO as no-drug con-trol. About 100 µl of cells were collected at specified time intervals, serially diluted in PBS ($10^0$ to $10^{-9}$ dilutions) and spread onto blood agar plates. Cell viability was determined by counting colony-forming units (CFUs).

### *A. baumannii* time-lapse imaging
Cells were grown overnight, diluted to an $OD_{600nm}$ of 0.01 and grown for 3 h at 37 °C, as described in the 'Growth conditions' section. Cells

were then spotted on MH II + 1% agarose pads, supplemented or not with 8 µg/ml of nitroxoline (fourfold MIC) between a glass slide and a coverslip. Slides were sealed with Valap to avoid coverslip shifting. Imaging was performed at room temperature every 15 min for 10 h over three distinct points of the slides for each condition using a Nikon Eclipse Ti inverted microscope, equipped with a Nikon DS-Qi2 camera, a Nikon Plan Apo Lambda ×60 oil Ph3 DM phase-contrast objective. Images were acquired with NIS-Elements AR4.50.00 software and processed with Fiji v.2.9.0/1.53t.

### Gentamicin protection assay and quantification of intracellular *Salmonella*
HeLa cells (ACC 57, German Collection of Microorganisms and Cell Cultures, Braunschweig, Germany) were cultivated in cell culture flasks (75 cm²) with RPMI medium. One day prior to infection, cells were seeded into 24-well plates ($10^5$ cells per well). *S.* Typhi clinical isolates (Supplementary Data 2) were grown overnight in LB medium and subcultured (1:33) for three h at 37 °C with shaking. Bacteria were harvested ($13,000 \times g$, 5 min), resuspended in RPMI medium and sub-sequently used for infection at a multiplicity of infection (MOI) 100 for 10 min. After discarding the supernatant, cells were washed with PBS, incubated for 40 min in RPMI with 100 µg/ml gentamicin to eliminate extracellular bacteria. Cells were then washed twice with PBS and incubated with RPMI with 5 µg/ml nitroxoline (previously tested to exclude toxicity on HeLa cells alone) or DMSO as vehicle control for 7 h. Infected cells were washed with PBS, lysed with 1% Triton-X and 0.1% SDS, and serial dilutions in PBS ($10^0$–$10^{-3}$) were plated on LB agar. Plates were incubated overnight at 37 °C and bacterial colonies were counted in at least four independent experiments in two technical replicates. Toxicity to HeLa cells was verified for seven nitroxoline concentrations along a twofold dilution gradient above and below the dose tested here (5 µg/ml) and for the Triton-X and medium controls (Supplementary Fig. 9).

### Concentration pretesting for checkerboard assay and data analysis
Twelve twofold serial dilutions of nitroxoline and 32 other compounds (Supplementary Data 1) were arrayed in technical duplicates in 384-well plates (ref. 781271 by Greiner BioOne) and inoculated with *E. coli* K-12 at a starting $OD_{600nm}$ of 0.001. Plates were sealed with breathable membranes incubated at 37 °C with continuous shaking and $OD_{600nm}$ was measured every 30 min for 14 h. The background, corresponding to the $OD_{600nm}$ at the first time point, was subtracted from each measurement for each well. The point at the transition from expo-nential to stationary phase was detected for each well, and the corre-sponding $OD_{600nm}$ was normalised by the median of the corresponding value of the no-drug controls present in each plate ($n = 16$). For each drug, the $IC_{75}$, i.e. the concentration at which 75% of the growth was inhibited, was identified in the resulting dose-response curves for each drug. For the checkerboard assay, eight evenly spaced concentrations were then selected, with the highest one correspond-ing to the $IC_{75}$ and the lowest one corresponding to the no-drug con-trol. Nitroxoline was tested in almost all combinations at $IC_{50}$ to be ideally placed to discover synergies as potential combinatorial regi-mens. All experiments were conducted in biological duplicates (i.e. plates inoculated with overnight cultures from distinct colonies).

### Checkerboard assay and data analysis
Pairwise combinations of nitroxoline with 32 compounds (Supple-mentary Data 1) were tested in a checkerboard microdilution assay. Drugs were arrayed in 8 × 8 checkerboards using the eight con-centrations previously selected. Growth was measured in the same conditions and data were analysed as for the concentration pretesting. The $OD_{600nm}$ at the transition between the exponential and stationary phase, after background subtraction, was normalised by the median of

the corresponding value of the no-drug controls present in each plate ($n = 6$). This value was used to calculate Bliss interaction scores[80] ($\varepsilon$) for each drug-drug concentration combination as follows:

$$\varepsilon = f_{d1,d2} - f_{d1} * f_{d2} \qquad (1)$$

where $f_{d1,d2}$ corresponds to the observed fitness in the presence of the drug combination, and $f_{d1}$ and $f_{d2}$ correspond to the fitness in the presence of the two single drugs. We, therefore, obtained 49 $\varepsilon$ scores for each checkerboard replicate. All experiments were conducted in at least two biological replicates, resulting in at least 98 $\varepsilon$ scores for each combination. Synergies and antagonisms were assigned when the first and third quartile of the $\varepsilon$ distribution, respectively, exceeded |0.1| and the median $\varepsilon$ exceeded |0.03|. Cumulative Bliss scores for each combination were considered as first quartile, third quartile and median, for synergies, antagonisms and neutral interactions, respectively. The Bliss interaction model was chosen as most suited for our screen design, where we tested three concentrations to maintain throughput and $IC_{50}$ or $IC_{75}$ as the highest doses to discover synergies, i.e. not mapping the full dose-response space for all drugs, which is required for other interaction models (e.g. Loewe model)[81].

### Resensitization of colistin-resistant strains by nitroxoline

Cells were pre-cultured as in 'Growth conditions', growth was measured and data were analysed as in 'Concentration pretesting for checkerboard assay and data analysis' in plates containing eight two-fold dilutions of colistin, supplemented or not with nitroxoline at 0.75 µg/ml. The strains used are listed in Supplementary Data 2.

### Two-dimensional thermal proteome profiling (2D-TPP)

Cells were grown overnight, diluted 1000-fold and grown until $OD_{578nm}$ ~0.6 at 37 °C, as described in the 'Growth conditions' section. After the addition of nitroxoline at the selected concentrations (0.02, 0.08, 0.4 and 2 µg/ml) or a vehicle-treated control, cultures were incubated at 37 °C for 10 min ($n = 1$ for each concentration). After $4000 \times g$ centrifugation for 5 min, cells were washed with 10 ml PBS containing the drug at the appropriate concentrations and resuspended in the same buffer to an $OD_{578nm}$ of 10. About 100 µl of this suspension was then aliquoted to ten wells of a PCR plate that was centrifuged at $4000 \times g$ for 5 min. About 80 µl of the supernatant was removed before exposing the plate to a temperature gradient for 3 min in a PCR machine (Agilent SureCycler 8800), followed by 3 min at room temperature. Cells were lysed with 30 µl lysis buffer (final concentration: 50 µg/ml lysozyme, 0.8% NP-40, 1x protease inhibitor (Roche), 250 U/ml benzonase and 1 mM $MgCl_2$ in PBS) for 20 min, shaking at room temperature, followed by three freeze-thaw cycles. Protein aggregates were removed by centrifuging the plate at $2000 \times g$ and filtering the supernatant at $500 \times g$ through a 0.45 µm filter plate (Millipore, ref: MSHVN4550) for 5 min at 4 °C. Proteins were then quantified with MS-based proteomics as previously described[73]. Briefly, proteins (2 µg) were digested according to a modified SP3 protocol[82], by adding them to the bead suspension (10 µg of Sera-Mag Speed Beads, Thermo Fisher Scientific) in 10 µl 15% formic acid and 30 µl ethanol), and after a 15 min incubation at room temperature with shaking, washing them four times with 70% ethanol. Proteins were digested overnight by adding 40 µl of 200 ng trypsin, 200 ng LysC, 5 mM chloroacetamide and 1.25 mM TCEP in 100 mM HEPES pH 8. Peptides were then eluted from the beads, dried under vacuum, reconstituted in 10 µl of water, and labelled for 1 h at room temperature with 17 µg of TMT10plex (Thermo Fisher Scientific) dissolved in 4 µl of acetonitrile. The reaction was quenched with 4 µl of 5% hydroxylamine. Experiments belonging to the same mass spectrometry run were combined. Samples were desalted with solid-phase extraction by loading the samples onto a Waters OASIS HLB µElution Plate (30 µm), washing them twice with 100 µl of 0.05% formic acid,

eluting them with 100 µl of 80% acetonitrile, and drying them under vacuum. Finally, samples were fractionated onto six final fractions on a reversed-phase C18 system running under high pH conditions (mobile phase A: 20 mM ammonium formate (pH 10) and mobile phase B: acetonitrile). Samples were analysed with liquid chromatography (UltiMate 3000 RSLCnano system) coupled to tandem mass spectrometry (Q Exactive Plus), with peptides separated in a Waters nanoEase HSS C18 T3 column (75 µm × 25 cm, 1.8 µm, 100 Å) with a 90 min gradient (mobile phase A: 0.1% formic acid in LC–MS grade water and mobile phase B: 0.1% formic acid in LC–MS grade acetonitrile). The mass spectrometer was operated in positive ion mode in data-dependent acquisition mode, with the top ten peptides being fragmented.

### 2D-TPP data analysis

Proteins were identified by searching the data against the *E. coli* K-12 strain Uniprot FASTA (Proteome ID: UP000000625), modified to include known contaminants and the reversed protein sequences, using Mascot 2.4 (Matrix Science) and isobarQuant[83]. Data analysis was performed using the R package TPP2D[84]. In brief, a null model, assuming that the soluble protein fraction depends only on temperature, and an alternative model, assuming a sigmoidal dose-response function for each temperature tested, were fitted to the data. For each protein, an F-statistic was obtained from the comparison of the residual sum of squares (RSS) of the two models. The abundance or thermal stability effect size were calculated for each protein as:

$$\text{sign}(k) \cdot \sqrt{RSS^0 - RSS^1} \qquad (2)$$

where k is the slope of the dose-response model fitted across temperatures and drug concentrations, $RSS^0$ and $RSS^1$ correspond to the residual sum of squares of the null (pEC50 linearly scaling with temperature) and alternative model, respectively[84].

### Gene Ontology (GO) enrichment

The enrichment analysis was performed on proteomes of *E. coli* BW251113 for the 2D-TPP data and for the proteomics data, and of the strains listed in Supplementary Data 2 as used for 'nitroxoline resistance evolution' for the analysis of proteomics data on sensitive and resistant strains. Proteomes were annotated using GOs downloaded from http://geneontology.org/ (release 2022-11-03). For each GO term, the enrichment of input protein sets (hits corresponding to FDR <0.05) against the background (all detected proteins) was tested using Fisher's exact test. *P* values were corrected for multiple testing using the Benjamini–Hochberg procedure.

### Nitroxoline MIC in *E. coli lptD4213*

Cells were grown as in 'Growth conditions'. Growth was measured as in 'Concentration pretesting for checkerboard assay and data analysis' upon exposure to seven nitroxoline twofold dilutions in *E. coli* BW25113 and *E. coli lptD4213*. Data was analysed as in 'Resensitization of colistin-resistant strains by nitroxoline'. Experiments were conducted in three biological replicates.

### Evaluation of drug combination therapy using the *G. mellonella* infection model

Larvae of the greater wax moth (*Galleria mellonella*) were infected with *Klebsiella pneumoniae* and treated with single drugs or drug combinations as previously described[85]. Caterpillars were purchased from Valomolia (Strasbourg, France). Stock solutions of colistin and nitroxoline were freshly prepared with 20 mM sodium acetate buffer (pH 5). Drug toxicity was preliminarily determined by injecting larvae with serial dilutions of single drugs and combinations. Non-toxic concentrations of the drugs were used for further experiments. Bacterial overnight cultures (grown as described in 'Growth conditions')

were diluted 1:100 with fresh MH II broth and incubated at 37 °C with continuous shaking, until an $OD_{600nm}$ of 0.2 was reached. Bacteria were then washed with PBS and adjusted to $5.6 \times 10^7$ CFU/ml, corresponding to a median lethal dose of 60–70% after 24 h, as determined in preliminary experiments. Groups of 10 caterpillars per condition were injected with 10 μl of the bacterial suspension into the haemocoel via the last right proleg and incubated at 37 °C. One hour post-infection, caterpillars were injected into the last left proleg with 10 μl of single drugs or drug combinations (0.1 μg/ml colistin and 0.1 μg/ml nitroxoline). Survival was monitored for 72 h. Each strain–drug combination was evaluated in three independent experiments. The statistical analysis was performed using the log-rank test.

## Efficiency of plating (EOP) assay
Cells were grown overday for 8 h as in 'Growth conditions' and tenfold serially diluted eight times. From each dilution, 3 μl were spotted onto MH II plates supplemented or not with 0.8 mM EDTA–0.5% SDS, 0.45 μg/ml nitroxoline, and a combination of the two conditions. Spots were allowed to dry and the plates were incubated overnight at 37 °C. Experiments were conducted in four biological replicates for each condition.

## N-phenylnaphthylamine (NPN)-fluorescence assay for OM damage
The assay was conducted as previously described[86]. Briefly, cells were grown overnight as described in the 'Growth conditions' section and diluted to an $OD_{600nm} = 0.5$ in 5 mM pH 7.2 HEPES buffer (Sigma Aldrich). About 100 μl of the cell suspension, together with 50 μl drugs diluted in HEPES buffer at the appropriate concentrations and 50 μl N-phenyl-1-naphthylamine (NPN) diluted in HEPES to a final concentration of 10 μM, were added to a black 96-well plate with clear-bottomed wells. Controls included all possible combinations of cells, drugs and NPN, each of them separately, and a plain buffer control. Fluorescence was measured immediately on a Tecan Safire2 plate reader using an excitation wavelength of 355 nm and an emission wavelength of 405 nm. Fluorescence measurements were obtained every 30 s for 10 min. After averaging across the 20 replicated measurements, the NPN Uptake Factor was calculated as follows:

$$\frac{Fluorescence_{drug + cells + NPN} - Fluorescence_{drug + cells - NPN}}{Fluorescence_{drug - cells + NPN} - Fluorescence_{drug - cells - NPN}} \quad (3)$$

Finally, the uptake values of samples containing drugs were compared to the no-drug control. Positive controls included 10 μg/ml polymyxin B and 0.4 mM EDTA. As a negative control, a non-OM-perturbing antibiotic (chloramphenicol) at its MIC (8 μg/ml) was included. Because of quenching between the NPN emission wavelength and nitroxoline excitation wavelength, nitroxoline concentrations higher than 2 μg/ml showed a linear decrease in fluorescence and were not used. Experiments were conducted in four biological replicates.

## Measurement of metal abundance and distribution via synchrotron radiation-induced X-ray fluorescence nano-imaging
Experiments were performed at the Nano-imaging beamline ID16A of the European Synchrotron Radiation Facility (ESRF). E. coli BW25113 cells were grown overnight in LB as described in 'Growth conditions', subcultured until reaching $OD_{600nm}$ 0.1 and treated for 15 min with nitroxoline (1 μg/ml) or 0.1% DMSO at 37 °C. After washing twice in PBS, 10 μl were mounted on silicon nitride membranes (Silson, Southam, UK, 1.5 mm × 1.5 mm × 0.5 μm) before cryo-fixation using a freeze plunger (EM GP, Leica) with 1 s blotting time. A 17 keV X-ray beam was focused to a 45 nm horizontal × 37 nm vertical spot by a pair of multilayer-coated Kirkpatrick–Baez mirrors located 185 m downstream of the undulator source with a high flux of $4.1 \times 10^{11}$ ph/s. The samples were rostered through the focal spot of the beam under a

vacuum of $10^{-7}$ mbar at −179 °C. XRF spectra were measured using a pair of element silicon drift diode detectors (7-element detector Vortex-ME7, Hitachi, and 16-element detector) to subsequently quantify elements by their K-level emission lines. Low-resolution and fast-position mapping by combined X-ray phase contrast and XRF coarse scans in low-dose mode ($6.1 \times 10^{10}$ ph/s) were performed using a scan step size of $300 \times 300$ nm$^2$ and a dwell time of 100 ms to identify bacteria. XRF fine scans in high-dose mode ($2.49 \times 10^{11}$ ph/s) were then performed with a step size of either $30 \times 30$ or $15 \times 15$ nm$^2$ and a dwell time of 50 ms to obtain quantitative elemental density maps from individual point spectra. After fitting and normalising the data with PyMca XRF spectral analysis software, mean intracellular elemental area density (ng/mm$^2$) were calculated from at least two different areas from two independent experiments.

## E. coli chemical genetic screen
Nitroxoline was tested on the E. coli whole-genome single-gene deletion mutant Keio collection[43] as previously described[42]. The collection (two independent clones per mutant), which was cryopreserved in 384-well plates, was arrayed in a 1536-colony format using a Rotor HDA (Singer Instruments). Cells were grown at 37 °C for 10 h and pinned on LB plates, with or without nitroxoline (2 μg/ml), in three replicates. After 16 h of incubation at 37 °C, plates were imaged using a controlled-light setup (spImager, S&P Robotics) and an 18-megapixel Canon EOS Rebel T3i camera. Mutant growth was calculated by quantifying colony opacity, estimated with the Iris software[87]. To account for better growth at the edges of a plate, two outermost columns/rows were multiplicatively adjusted to the median opacity of the plate[88]. Mutant fitness was then estimated as a fraction of the plate median opacity. A change in mutant fitness was quantified as a multiplicative change per condition using a two-sided unpaired t-test. The resulting t-statistic was empirical Bayes' moderated[89] and corresponding p values were adjusted for multiple testing (Benjamini–Hochberg correction[90]) (Supplementary Data 4).

## Experimental resistance evolution
Nitroxoline-sensitive clinical isolates and reference strains of E. coli, K. pneumoniae and A. baumannii (Supplementary Data 2) were exposed to increasing nitroxoline concentrations from 0.5x MIC to 4x MIC (twofold dilution steps) in LB as previously described[29]. A defined bacterial inoculum (McFarland 0.5, corresponding to $10^8$ bacteria/ml) was passaged every 24 h for at least 7 days in 5 ml LB. In the case of bacterial growth, nitroxoline concentration was increased twofold. The MIC of the strains was measured using broth microdilution and agar dilution, as described before.

## Frequency of resistance
Single colonies from two isolates of A. baumannii, E. coli and K. pneumoniae were inoculated in 20 ml of MH II broth and incubated overnight at 37 °C and 180 rpm. Bacterial cells were then harvested by centrifugation and resuspended in 1 ml of MH II broth. Tenfold serial dilutions were then spread (100 μl) onto MH II agar plates with and without nitroxoline (4x MIC; 8xMIC) and incubated for 48 h at 37 °C. The mutation frequencies were then determined by dividing the number of CFUs on nitroxoline-supplemented agar by the number of CFUs on antibiotic-free agar. Experiments were performed from at least four independent experiments for each isolate. Resistance of colonies growing on nitroxoline-supplemented plates was confirmed by MIC determination as described above.

## Genome sequencing and single nucleotide polymorphism (SNP) analysis
Experimentally evolved strains were defined as resistant if the MIC fold increased by ≥4 compared to the parental strain. Clinical isolates were considered resistant if their MIC was at least two times the median

species MIC measured in this study (Fig. 1c and Supplementary Fig. 1a, b). Genome sequencing was performed for all isolates using short-read technology (MiSeq platform; Illumina, San Diego, CA), generating 150 or 250 bp paired-end reads and >100-fold average coverage. After quality trimming of the reads, de novo assembly and scaffolding was conducted using SPAdes version 3.12.0 with standard parameters. Annotation was done with Prokka version 1.14.6[91]. SNP analysis was performed using snippy (https://github.com/tseemann/snippy) to compare isogenic nitroxoline susceptible and resistant strains. Deletions were analysed using an in-house script. Clinical isolates were further compared to annotated reference genomes (GCF_000258865.1, GCF_000750555.1 and Bioproject PRJNA901493) to determine mutations in genes encoding for efflux pump regulators with an in-house script.

### Structural alignment and annotation of OqxR, EmrR, AdeL, AcrR/TetR

Amino acid changes in mutated proteins from nitroxoline-resistant clinical isolates were mapped onto protein features extracted from Proteins API[92] using a custom-script, adapted from the software mutplot[93] (UniProt IDs used: EmrR: P0ACR9; AdeL: A0A059ZJX1; AcrR/TetR: A0A245ZZS0). Because OqxR in *K. pneumoniae* lacks a UniProt ID and feature annotation, we searched for its closest, feature-annotated, structural homologue with Foldseek using the 3Di/AA mode[94]. The highest-ranking protein (E-value 7.58E-07, score 265) with available domain annotation was another Rrf2 transcription factor, NsrR, from *S. enterica subsp. enterica* serovar Typhimurium LT2 (AF-Q8ZKA3-F1-model_v4). Structures were visualised using Mol* Viewer[95] from PDB[96]. Alignment of wild-type and mutated OqxR structures were performed using the Pairwise Structure Alignment tool on RCSB PDB[97].

### Complementation of nitroxoline-resistant isolates

Clinical isolates with *emrR* or *oqxR* mutations were transformed with pTOPO expression plasmids (pCR-Blunt II-TOPO, Invitrogen) harbouring the wild-type gene (pTOPO_*emrR* or pTOPO_*oqxR*) via electroporation as previously described[98]. For this purpose, bacteria were grown overnight as described in "Growth conditions". On the next day, cells were subcultivated (1:100 dilution) until an $OD_{600nm}$ of 0.4–0.6 was reached. Cells were harvested (13,000 × g, 3 min), washed once in 500 µl 300 mM ice-cold sucrose, and transformed with 500 ng of plasmid DNA with a Gene Pulser Xcell electroporator (Bio-Rad) with 2.2 kV, 200 Ω and 25-µF settings. Cells were recovered in SOC medium at 37 °C for 1 h (with shaking at 180 rpm) and plated on LB agar plates with kanamycin (30 mg/l for *E. coli* and 100 mg/l for *K. pneumoniae* and *A. baumannii*) for selection of transformants, subsequently used for MIC testing.

### Proteomics of resistant strains

Cells were grown overnight as described in 'Growth conditions', diluted to $OD_{600nm}$ = 0.05 in 3 ml LB, grown until reaching $OD_{600nm}$ = 0.5. Nitroxoline-sensitive strains were treated with 1x MIC nitroxoline (0.5–4 µg/ml depending on the strain) or with the DMSO control for 10 min. Nitroxoline-resistant strains were exposed only to DMSO. After 4000 × g centrifugation for 5 min, 2 ml aliquots were washed with 1 ml PBS (containing the drug at the appropriate concentration for drug-exposed samples). The final pellets were frozen at −20 °C until analysis, when they were resuspended in lysis buffer (final concentration: 2% SDS, 250 U/ml benzonase and 1 mM $MgCl_2$ in PBS) and immediately incubated at 99 °C for 10 min. Protein digestion, peptide labelling, and MS-based proteomics were performed as described above for 2D-TPP[41]. Limma analysis was performed similarly as previously described[41] to determine proteins that were significantly up or downregulated.

### Orthology analysis of resistant strains

All complete genomes belonging to the *A. baumannii*, *E. coli* and *K. pneumoniae* species were downloaded from NCBI RefSeq using ncbi-genome-download[99]. Newly sequenced genomes were annotated using prokka, with default parameters[91]. The pangenome for each of the three species was computed separately using panaroo with the '--clean-mode strict --merge_paralogs' options[100]. One or more protein sequences for each gene were then sampled in the three pangenomes, giving priority to parental-sensitive strains. If a gene was not present in any of these 'focal' strains, a random strain was selected. Sampled protein sequences were annotated using eggnog-mapper, with the following parameters: '--target_orthologs one2one --go_evidence all --tax_scope Bacteria --pfam_realign realign'[101]. GO terms associated with each gene cluster were recovered using this automatic annotation. We further expanded this set by querying the NCBI protein database using Biopython's Entrez interface[102]. This was possible because we used complete genomes from RefSeq. We then combined the two annotation sets to derive a set of GO terms for each gene cluster in the three species.

### Reporting summary

Further information on research design is available in the Nature Portfolio Reporting Summary linked to this article.

## Data availability

Source data for all figures are provided with this paper. The mass spectrometry proteomics data have been deposited to the ProteomeXchange Consortium via the PRIDE partner repository with the dataset identifiers PXD050778 and PXD050827. Genome sequences of strains used for resistance evolution can be found under NCBI Bio-Project ID PRJNA1194642. The chemical genetic data generated in this study can be found at https://github.com/ElisabettaCacace/nitroxoline_2024. Source data are provided with this paper.

## Code availability

All code used for the analysis and figure generation is available at https://github.com/ElisabettaCacace/nitroxoline_2024.

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

## Acknowledgements

S.G. was supported by the Rolf. M. Schwiete-Stiftung. A.T., V.V., F.C., A.B.-N. and M.Z. were supported by EMBL. M.K. was supported by Vetenskapsrådet 2019-00666. A.M. was supported by a fellowship from the EMBL Interdisciplinary Postdoc (EI3POD) programme under Marie Skłodowska-Curie Actions COFUND (grant number 664726). M.G. was funded by the Deutsche Forschungsgemeinschaft (DFG, German Research Foundation) under Germany's Excellence Strategy—EXC 2155—project number 390874280. A.O. and P.B. were supported by the German Federal Ministry of Education and Research (LAMarCK, 031L0181A to P.B.). M.M.S. was supported by the Allen Distinguished Investigator Award through the Paul G. Allen Frontiers Group". We thank ESRF for granting beamtime on ID16A through experiment LS –3269 (DOI 10.15151/ESRF-ES-1346211120). We are thankful to Franka-Maria Schubert, Anna-Lena Hesse, Anna Maldener and Bettina Stietz for excellent technical support.

## Author contributions

E.C. and S.G. conceived and designed the study. S.G. and A.T. supervised the study. E.C., M.T., M.S. and S.G. designed the experiments. M.S. and S.G. performed and analysed the MIC testing of clinical isolates; M.T., M.S. and J.P. analysed resistant mutants. M.T. and J.P. performed time-kill experiments and *S.* Typhi intracellular killing assay. M.T. and M.S. performed efflux pump inhibitor experiments. M.S. performed nitroxoline resistance evolution and sequencing of resistant clones. M.T. performed *G. mellonella* infection experiments. M.T., M.E., C.R., P.C. and S.G. performed the XRF experiment. E.C. performed and analysed the time-lapse microscopy, drug pretesting and combination screen with the help of M.K., *E. coli lptD4213* MIC testing, experiments on colistin-resistant *Enterobacteriaceae*, EOP and NPN assays with the help of A.B.-N. and input from M.Z. E.C. and A.O. performed the phylogeny analysis with input from P.B. A.M. performed the 2D-TPP and proteomics experiments, and E.C. analysed the data, with input from M.M.S. M.G. performed the orthology analysis of resistant strains. A.K. and F.C. performed the chemical genetic pretesting and screen; V.V. analysed the data. T.G.S. analysed the genome sequences and performed the SNP analysis. E.C. and S.G. wrote the manuscript with input from all authors. All authors approved the final version.

## Funding

## Competing interests

The authors declare no competing interests.
