## [Transparent Peer Review file · Nature Communications]

Uncovering nitroxoline activity spectrum, mode of action and resistance across Gram-negative bacteria

Corresponding Author: Professor Stephan Goettig

Version 0:

Reviewer comments:

Reviewer #1

(Remarks to the Author)

Cacace and Tietgen et al., explore the spectrum of activity of the antibiotic nitroxoline (NTX), which has been used clinically for >50 years to treat uncomplicated UTIs caused by *E. coli* or *K. pneumoniae* in Europe. Through large-scale susceptibility testing of >1000 clinical isolates, they demonstrate the antibiotic exhibits promising broad-spectrum activity against Gram-negative bacteria. They also identify drug interactions that could enhance, or negatively impact, antibacterial activity. Despite this antibiotic being in clinical use for some time, the mechanism of action (MOA) is surprisingly not well understood. The authors undertook a systems approach to better understand the MOA, revealing that NTX disrupts the outer membrane and may act as a metallophore, inducing metal toxicity. Finally, they also experimentally evolved resistance to NTX, revealing increased efflux activity, in line with other published work.

Overall, this work provides important information relating to a clinically used antibiotic. Despite being used for >50 years, NTX remains efficacious for the treatment of UTIs, with low resistance rates. The authors work reveals NTX could also be used to treat infections caused by myriad other pathogenic Gram-negative bacteria, for which treatment options are lacking in the face of increased resistance to other antibacterial agents. The proposed MOA could explain why NTX resistance is slow to emerge, further supporting the use of NTX for the treatment of infections caused by Gram-negative bacteria. Additionally, the prospect of using NTX as an antibiotic adjuvant is particularly compelling. Thus, I believe the work could be of significance to the field.

Despite my enthusiasm for the work, there are limitations, much of which are concerned with data presentation and improvements to the manuscript text. Importantly, there is not enough detail in the methods for the work to be reproduced, especially the intricate systems work, with reference to previous studies throughout, which made it difficult to assess the appropriateness of the methodology and the interpretation of the data.

I found many of the figures to be counterintuitive and the figure legends too long with methodology and not enough information relating to statistics, nature of replicates and sample size in some instances.

The introduction should be more detailed, and it is contradictory in places; it is stated NTX has activity against a broad spectrum of Gram-negative bacteria (Line 52), but then in line 68 the authors propose it has broader activity than believed, but it is unclear what was known previously. Reference 56 should be included in the introduction to provide some more context, and others as necessary.

NTX displays activity against a variety of microbes, including fungi, and also has other potential uses (e.g., anticancer). It is surprising that it is not toxic to human cells, could the authors comment on this?

The cidal vs static aspect of the work is interesting but could be expanded on in the discussion; why is it cidal in some organisms and not others. How extensively was this explored?

The large-scale activity profiling (Figure 1) reveals broad-spectrum activity against Gram-negative bacteria; however, I was surprised to see a lack of Gram-positive pathogens. Additionally, it would be advisable to include the MIC90 for each species.

The antibiotic combinatorial work is a strength; however, the fractional inhibitory concentration index (FICI) should be calculated and listed for each checkerboard.

The proposed MOA could be clearer in the abstract (Lines 362-364 were helpful). It is not obvious to me whether OM disruption or metal intoxication is the major contributor to NTX's MOA. Is the antibiotic removing cations from the OM, and in this instance is acting as a chelator?

The rationale for using 2D-TPP is not well explained and the data not easy to interpret in the figures. The use of permeabilized strains adds a layer of complexity since this could be allowing a greater concentration of NTX into the cell. It is unclear how the authors believe NTX is acting on the OM, and how this relates to the LptD mutant.

The authors describe RNAP inhibition but did not explore this in their study. A coupled transcription/translation assay should be used, perhaps this too is another contributor to antibacterial activity?

Finally, the resistance evolution study was appreciated, and confirmed previous findings (Reference 65); however, it is unusual that the authors did not include the frequency of resistance/mutation (FOR/FOM), which would be critical for advancing NTX for other indications.

Other comments and proposed edits:

- In the introduction, NTX background could be expanded.
- Very long figure legends including much methodology.
- provide first in-text reference to acronyms (e.g., OM, Line 73)

Figure 1: Could be improved. Figure 1b is confusing. Figure 1d suggests NTX is static intracellular and doesn't eradicate. Was killing determined for this organism? Is there a reason the MIC90 was not determined for each species and/or genera? The key on panel c should be relocated.

Figure 2 & ext Fig.3: Figure 2a: FICI would be preferable since this is the standard in the field of combinations. Figure S3: which organism was used in this experiment? 'One of the most potent synergies of nitroxoline was with the OM-targeting drug colistin', requires validation, colistin does not look synergistic, FICI should be calculated. 2-fold is within the standard error range of these experiments. The drug concentration range should have been adjusted since many antibiotics did not inhibit growth alone, and therefore interactions could have been missed. The grey scale is difficult to assess. There seems to be synergy with piperacillin, rather than antagonism like the other beta-lactams? Antagonism is noteworthy, and worth highlighting to the clinical field, yet this is not well emphasized in the manuscript. Figure 2 c, statistics are not shown. Suggest removing 'animal'. Panel b is confusing. The authors could include a single strain as an example and move the others to the supplementary. Why was this measured at 10.75 h growth?

Figure 3: Panel a is poorly descriptive. Is this looking for changes in abundance or Tm differences? Nature of replicates etc not included.

Method detail could be improved throughout, avoiding reference to previous studies, which makes the manuscript very difficult to review. E.g., Line 487: Method detail is lacking, nature of replicates is essential. It is unclear how the samples were measured.

Figure 5 – panel e, the scale is not appropriate, and it is difficult to interpret the outcomes of the experiment.

Line 62 – in the clinic?

Line 251 – What is the frequency of resistance?

Line 538: bacteria were grown ON and harvested at 0.2?

Line 605 – move this section after the proteomics.

Reviewer #2

(Remarks to the Author)

The authors of the manuscript "Uncovering nitroxoline activity spectrum, mode of action and resistance across Gram-negative bacteria" have conducted an impressive study. This manuscript examines the broad-spectrum applicability of the quinolone antibiotic, nitroxoline. The authors evaluated the in vitro efficacy of nitroxoline against a diverse and large panel of pathogens, including multidrug-resistant (MDR) clinical bacterial strains. They demonstrated that nitroxoline possesses species-specific bactericidal activity and utilized systems-based approaches to elucidate its antibacterial mode of action. Additionally, the study explores the development of resistance to nitroxoline, primarily through efflux mechanisms, across various Gram-negative bacterial species.

However, in the context of existing literature, there appears to be considerable overlap with previously established knowledge on nitroxoline. Although the authors have conducted commendable work, particularly considering the sample size and the application of systems-biology approaches, the novelty of the study is limited.

Firstly, the antibacterial activity spectrum of nitroxoline has already been reported against various bacterial pathogens beyond *E. coli*, such as *K. pneumoniae*, *P. aeruginosa*, *A. baumannii*, *S. aureus*, and *Enterococci* sp. (References: PMID: 31423123, 32899218, 32364820, 31423123, 31292653). The species-specific bactericidal nature of nitroxoline has also been demonstrated against other bacteria (Reference: PMC6395908). Additionally, the metal chelation activity of nitroxoline against zinc and copper has been previously documented (References: PMID: 31423123, 22926564). The ability of nitroxoline to disrupt the outer membrane has also been shown (Reference: PMID: 38085067). Moreover, the role of RND efflux pumps in resistance development against nitroxoline has been shown in literature as has also been discussed in the manuscript (PMID: 38063385; 31633764).

Reviewer #3

(Remarks to the Author)

In this manuscript the authors interrogate the mechanism of action of the antibiotic nitroxoline. The authors set out to test if nitroxoline could be repurposed against a wide range of bacterial pathogens to expand its therapeutic profile. These results revealed a promising antimicrobial profile against Enterobacteriales, Acinetobacter, and intracellular Salmonella. The authors also report the surprising finding that nitroxoline is bactericidal in certain cases and that it potentiates the therapeutic benefit of colistin in a Galleria model of Klebsiella infection. Using a chemical genetics approach, the authors suggest that nitroxoline affects LPS biosynthesis. In addition, nitroxoline exposure led to an increase in the abundance of metallothioneins and regulators which are involved in metal homeostasis, suggesting a role for nitroxoline in metal intoxication. The authors conclude the manuscript with identification of mutations that provide nitroxoline resistance in multiple bacterial species. The authors conclude that these findings suggest that nitroxoline does not have a protein target and that its therapeutic window is now wider for the treatment of infectious diseases. The manuscript is well written, the experiments are well controlled, and the results are provocative and will be of interest to a wide audience. I have only one question for the authors to consider.

1. It is unusual that metal import and efflux mutants are both more sensitive to nitroxoline. Can the authors provide text to explain this?

Version 1:

Reviewer comments:

Reviewer #1

(Remarks to the Author)

The authors have done an excellent job responding to the reviewers' comments, conducting additional experiments, and revising the manuscript to enhance its clarity and overall quality.

Open Access This Peer Review File is licensed under a Creative Commons Attribution 4.0 International License, which permits use, sharing, adaptation, distribution and reproduction in any medium or format, as long as you give appropriate credit to the original author(s) and the source, provide a link to the Creative Commons license, and indicate if changes were

made.

Point-by-point response to the referees (Cacace *et al.*, NCOMMS-24-36511)

We thank the reviewers for their feedback, which helped us to identify limitations and improve our manuscript. In this revised version we have addressed all main points raised, i.e. improving the documentation of systems approaches, clarifying our advancements compared to previous knowledge, and providing additional data on nitroxoline resistance potential. This resulted in the rephrasing of the main text, improvement of figures, an updated Source Data Table including species MIC₅₀ and MIC₉₀, the addition of one Supplementary Table with frequency of nitroxoline resistance and Supplementary Figures on nitroxoline toxicity to eukaryotic cells and colistin-nitroxoline synergy.

Reviewer #1:

Cacace and Tietgen *et al.*, explore the spectrum of activity of the antibiotic nitroxoline (nitroxoline), which has been used clinically for >50 years to treat uncomplicated UTIs caused by *E. coli* or *K. pneumoniae* in Europe. Through large-scale susceptibility testing of >1000 clinical isolates, they demonstrate the antibiotic exhibits promising broad-spectrum activity against Gram-negative bacteria. They also identify drug interactions that could enhance, or negatively impact, antibacterial activity. Despite this antibiotic being in clinical use for some time, the mechanism of action (MOA) is surprisingly not well understood. The authors undertook a systems approach to better understand the MOA, revealing that nitroxoline disrupts the outer membrane and may act as a metallophore, inducing metal toxicity. Finally, they also experimentally evolved resistance to nitroxoline, revealing increased efflux activity, in line with other published work.

Overall, this work provides important information relating to a clinically used antibiotic. Despite being used for > 50 years, nitroxoline remains efficacious for the treatment of UTIs, with low resistance rates. The authors work reveals nitroxoline could also be used to treat infections caused by myriad other pathogenic Gram-negative bacteria, for which treatment options are lacking in the face of increased resistance to other antibacterial agents. The proposed MOA could explain why nitroxoline resistance is slow to emerge, further supporting the use of nitroxoline for the treatment of infections caused by Gram-negative bacteria. Additionally, the prospect of using nitroxoline as an antibiotic adjuvant is particularly compelling. Thus, I believe the work could be of significance to the field.

1. Despite my enthusiasm for the work, there are limitations, much of which are concerned with data presentation and improvements to the manuscript text. Importantly, there is not enough detail in the methods for the work to be reproduced, especially the intricate systems work, with ref to previous studies throughout, which made it difficult to assess the appropriateness of the methodology and the interpretation of the data.

We thank the reviewer for the helpful comments. To ensure transparency and to improve method clarity, we expanded the Methods sections related to the systems approaches that we used (drug combination screen, chemical genetics, 2D-TPP and proteomics), rephrased the relevant figure captions and stored all data and code used for the analysis on public repositories (GitHub, ProteomeXchange Consortium, NCBI BioProject).

Action points: we expanded the Methods section related to drug combination screen (lines 515-519) and 2D-TPP (lines 541-566). We specified the number and nature of replicates in figure captions (lines 530, 864-865 for 2D-TPP, line 872 for chemical genetics). We added the

links of all public data repositories in the Data Availability section (lines 772-778). Additional explanations of our edits can be found in replies to points 2 and 9.

2. I found many of the figures to be counterintuitive and the figure legends too long with methodology and not enough information relating to statistics, nature of replicates and sample size in some instances.

Action points: we revised figures 1c, 2, 3a and 5a to enhance readability and removed non-essential information from the figure legends. Furthermore, we added helpful notes, statistical details and replicates/sample sizes in the figure legends as requested and where applicable (e.g. lines 530, 818-820, 864-867, 872, 887-888). Please see reviewer's point 9 and 13 for more details.

3. The introduction should be more detailed, and it is contradictory in places; it is stated nitroxoline has activity against a broad spectrum of Gram-negative bacteria (Line 52), but then in line 68 the authors propose it has broader activity than believed, but it is unclear what was known previously. Ref 56 should be included in the introduction to provide some more context, and others as necessary.

Action points: we expanded and rephrased the introduction (lines 53-73), anticipating ref. 56 (now ref. 30, PMID: 38063385), to specify nitroxoline's known activity spectrum and better highlight knowledge gaps addressed in the manuscript.

4. nitroxoline displays activity against a variety of microbes, including fungi, and also has other potential uses (e.g., anticancer). It is surprising that it is not toxic to human cells, could the authors comment on this?

The favorable safety profile of nitroxoline has been demonstrated through its use for more than 50 years with very few side effects and the usage as the first-line therapy for uncomplicated UTI (PMID: 29111434), indicating low toxicity to human cells *in vivo*. Nitroxoline's effects on human cells *in vitro* are variable: while many cancer cell types are affected via different mechanisms (ref. 14-17), non-cancerous cell types such as primary human fibroblasts and keratinocytes do not affect cell viability upon exposure of up to 20 μ M nitroxoline (PMID: 36897017).

While we did not explore in this manuscript nitroxoline's effects on eukaryotic cells or their cross-kingdom conservation, the mechanisms so far shown for nitroxoline anticancer activity seem to be orthogonal, host-specific processes: these include inhibition of metastasis via downregulation of metalloproteases (ref. 16), cathepsin inhibition (ref. 17) and disruption of the AMPK/mTOR pathway in prostate cancer cells, independent of nitroxoline's ion chelation properties (ref. 14).

Action points: we assessed and excluded nitroxoline toxicity on human cells for the intracellular *Salmonella* Typhi killing experiments (Fig. 1d). We added more information on this, specifying in the Methods that we performed a dimethylthiazol-2-yl-2,5-Diphenyltetrazolium Bromide (MTT)-based cell viability assay with Hela cells (lines 474-477). We added these results as a new Figure in the Supplementary File (Supplementary Fig. 1), showing an average cell viability of 92% at the nitroxoline dose of 5 μ g/mL used in the experiment (Fig. 1d).

5. The cidal vs static aspect of the work is interesting but could be expanded on in the discussion; why is it cidal in some organisms and not others. How extensively was this explored?

Increasing evidence indicates that bacteriostatic/bactericidal activity of antibiotics can be species- and even strain-specific (ref. 74). In our manuscript, we assessed killing only on individual *A. baumannii* and *E. coli* strains. Hence, these results cannot be generalized at the species level, as we underline in the discussion as a limitation of this study. As for the mechanisms of the bactericidal activity that we observed in *A. baumannii*, we deliberately limited ourselves to describing the microscopy results, with release of cytoplasmic content and lysis, at least in the tested strain 19606^T. This is likely caused by envelope damage, which should be investigated in detail in future studies.

Action points: we rephrased the discussion (lines 391-403), stressing that a separate and more extensive assessment of nitroxoline cidal activity on multiple strains per species and drug concentrations should be conducted in future to confirm this species-specificity and elucidate possible mechanisms.

6. The large-scale activity profiling (Figure 1) reveals broad-spectrum activity against Gram-negative bacteria; however, I was surprised to see a lack of Gram-positive pathogens. Additionally, it would be advisable to include the MIC₉₀ for each species.

We decided to focus on Gram-negative species as they represent the majority of the species listed in the WHO resistance priority list 2024 (ISBN: 978-92-4-009346-1), also considering that an extensive assessment of nitroxoline susceptibility for Gram-positive species has been previously performed (PMID: 29111434).

Action points: we determined MIC₅₀ and MIC₉₀ values for each species (see Source Data). We included MIC₅₀ values in Fig. 1c, since we refer to median MICs in the Results and Discussion. We updated the caption of Fig. 1c accordingly (lines 824-825). We adapted our statements in the Results about median species MIC, which corresponds to MIC₅₀ (lines 109, 115, 293). Of note, we only commented on species with at least 50 strains tested, where these cumulative estimates are most reliable (PMID: 20181573).

7. The antibiotic combinatorial work is a strength; however, the fractional inhibitory concentration index (FICI) should be calculated and listed for each checkerboard.

Bliss and Loewe models are the most used methods to quantify drug-drug interactions (PMID: 30323252). We are aware of each model's limitations and now explain the selection for Bliss in the Methods section. Loewe additivity, which FICI is based upon (PMIDs: 756833, 19995928 and 32941420), is inappropriate in our case, because it can only be calculated for checkerboard assays where the MIC is tested, i.e. the full drug dose-response is mapped.

In our checkerboards, we deliberately chose in almost all checkerboards sub-MIC concentrations for nitroxoline and many of the combined drugs, to have the opportunity to detect the most powerful synergies, effective even at low drug dosages. We hence calculated Bliss interaction scores, which can be accurately determined for sub-MIC concentrations, or for drugs that do not have activity of their own but can still synergize with other compounds. In the absence of an MIC in the same experiment for nitroxoline and other drugs in the screen, we would need to repeat the whole high-throughput screen including nitroxoline MIC in the

concentrations tested. We feel that this would add little value to the manuscript, especially considering that we could validate the synergy with colistin *in vivo* against colistin-resistant strains. We, however, repeated this synergy with concentration gradients that allow FICI calculation, confirming the synergy.

Action points: we expand our motivation for selecting drug concentrations and the Bliss interaction model in the Methods section (lines 515-519), introducing ref. 81 (PMID: 25756107). We retested and calculated FICI for the colistin-nitroxoline synergy (please see also graph in response to point 16).

8. The proposed MOA could be clearer in the abstract (Lines 362-364 were helpful). It is not obvious to me whether OM disruption or metal intoxication is the major contributor to nitroxoline's MOA. Is the antibiotic removing cations from the OM, and in this instance is acting as a chelator?

Our 2D-TPP data show that nitroxoline does not have a distinct protein target. Furthermore, our data do not indicate a prevailing mode of action for nitroxoline, but rather pleiotropic effects, including both OM disruption and copper and zinc intoxication. For both we show indirect and direct evidence. We comment on the possibility that nitroxoline disrupts LPS and OM stability by complexing Mg^{2+} and Ca^{2+} (lines 215-221, 372-377), referring to previous knowledge on the impact of cation concentration on nitroxoline MIC (ref. 22 and ref. 48). However, the changes that we detected via 2D-TPP included not only OMP destabilization, which could indeed be a consequence of direct OM disruption, but also decreased abundance and stability of components of LPS biosynthesis and trafficking (WaaC, GmhA, GmhB, Lpt machinery), which could also contribute to the observed OM damage in a potentially distinct, cation-independent way. Further studies will be required to tear apart causes and consequences of LPS synthesis, integrity and OM disruption and the impact of nitroxoline on these different aspects. We now expand and connect previous knowledge on nitroxoline chelation with our findings on metal intoxication and OM disruption in the Discussion.

Action points: we rephrased the abstract (lines 35-37) to clarify our findings on nitroxoline MoA. We rephrased the Results and cited PMID: 38085067 (ref. 48) as more recent evidence on nitroxoline's chelation (lines 215-221). We elaborate in the discussion (lines 372-387) our hypotheses on how our findings relate to current evidence on nitroxoline metal chelation, with particular attention to the species-specific differences that we observed in nitroxoline's antibacterial activity.

9. The rationale for using 2D-TPP is not well explained and the data not easy to interpret in the figures. The use of permeabilized strains adds a layer of complexity since this could be allowing a greater concentration of nitroxoline into the cell. It is unclear how the authors believe nitroxoline is acting on the OM, and how this relates to the LptD mutant.

Thermal proteome profiling (TPP) is an established method for an unbiased, proteome-wide search of direct and indirect drug targets (ref. 39-41). TPP is also the only method that attains this scope without requiring compound modifications and allows profiling of targets and off-target effects in living cells. This proved essential for nitroxoline, for which all hits were detected in living cells, with no significant changes in lysates (lines 177-181). Furthermore, the 2D format provides higher sensitivity across drug doses and the opportunity to assess both abundance and stability in the same experiment thanks to the temperature gradient. Altogether, these points led us to choose 2D-TPP as an ideal method for the investigation of nitroxoline's target

and to monitor stability and abundance of proteins upon nitroxoline exposure (Fig. 3a and Fig. 4a-c), which have not been investigated before.

Regarding the LptD mutant, 2D-TPP experiments were not performed in this mutant background. We assume the reviewer is referring to the experiment in ED Fig. 4d, which was performed as mere validation of the 2D-TPP (Fig. 3a) and chemical genetics (Fig. 3b) results, where LptD was a common hit. The LptD mutant is OM-defective and, hence, well suited to investigate nitroxoline's effect on OM as outlined in lines 206-213. We agree that the increased susceptibility to nitroxoline shown in ED Fig. 4d could also be due to increased permeability and intracellular nitroxoline concentration, and not to a synergistic perturbation of the OM. This is why we sought to provide more direct evidence of nitroxoline perturbation of the OM, i.e. the NPN assay (Fig. 3c) and the synergies with bulky antibiotics that could only get into the cell if the OM was disrupted (Fig. 2).

Action points: we expanded on the rationale for using 2D-TPP in the main text (lines 172-174) to highlight the specific advantages of 2D-TPP as a unique method to decouple direct drug interactions with targets and downstream effects. We added a paragraph with more information and references on this technique in the Methods section (lines 541-566), revised axes of Fig. 3 and rephrased the caption of Fig. 3 and Fig. 4 to improve figure readability.

10. The authors describe RNAP inhibition but did not explore this in their study. A coupled transcription/translation assay should be used, perhaps this too is another contributor to antibacterial activity?

Evidence of RNA polymerase inhibition has been provided 50 years ago in yeast for the nitroxoline analogue 8-hydroxyquinoline by Fraser *et al.* (ref. 1). The only evidence for inhibition of bacterial RNA polymerase is provided by the same authors *in vitro* (PMID: 810137; new ref. 27), using the same nitroxoline analogue, isolated *E. coli* RNA polymerase and yeast-derived RNA. Considering this, we believe that performing an *in vitro* transcription/translation assay as suggested would provide no new knowledge. Novel information might be provided by more elaborate approaches (e.g. structural characterization), which in our opinion would fall beyond the scope of our study and still leaves the open question whether such effects hold true *in vivo*.

Considering this and the reviewer's comment, we therefore realized that our statement in the abstract could be misleading for readers, suggesting RNA polymerase inhibition as a major demonstrated MoA for nitroxoline or even a main hypothesis/focus of our study.

Action point: we moved the description of RNA polymerase inhibition by nitroxoline from the abstract to the introduction (lines 61-63), summarizing the points we outlined above and adding PMID: 810137 as ref. 27 for this effect on bacteria.

11. Finally, the resistance evolution study was appreciated, and confirmed previous findings (Ref 65); however, it is unusual that the authors did not include the frequency of resistance/mutation (FOR/FOM), which would be critical for advancing nitroxoline for other indications.

We determined the frequency of resistance as requested for *E. coli*, *K. pneumoniae* and *A. baumannii* (two strains per species). The frequency of mutations was overall low for all species (mostly below 10^{-10}). For *E. coli* and *K. pneumoniae*, this is consistent with previous reports (ref. 30, PMID: 38063385), whereas *A. baumannii* hadn't been investigated before. This supports the point that nitroxoline could be repurposed to a much broader spectrum of Gram-negative species than currently.

Action points: we added the frequency of mutations for the three species (two strains per species, two tested MICs, four independent experiments) as Supplementary Table 5, and expanded the Results section (lines 273-278).

Other comments and proposed edits:

12. In the introduction, nitroxoline background could be expanded.

Action points: we expanded the background on nitroxoline known spectrum of activity and proposed mode(s) of action in the introduction as requested (lines 53-65). Please see also points 3 and 10 for more details.

13. Very long figure legends including much methodology.

Action points: we removed non-essential information from the figure legends. We integrated this suggestion with point 17, where the reviewer comments on the lack of methodological information for Fig. 3a, and 18, where it was suggested to add information on methods. We therefore moved more detailed methodological information from figure legends to the Methods and kept in the captions all information on statistics and number of replicates, as per journal guidelines and consistently with reviewer's point 2.

14. provide first in-text ref to acronyms (e.g., OM, Line 73)

Done as suggested.

15. Figure 1: Could be improved. Figure 1b is confusing. Figure 1d suggests nitroxoline is static intracellular and doesn't eradicate. Was killing determined for this organism? Is there a reason the MIC90 was not determined for each species and/or genera? The key on panel c should be relocated.

We thank the reviewer for these suggestions. For Figure 1b, we followed current indications of Nature Methods on how to represent sets and their intersections (<https://doi.org/10.1038/nmeth.3033>). We tried alternatives such as Venn diagram, but the data representation was not as clear and information got lost, e.g. on set size and importantly on how many species we tested with all three susceptibility methods. To address the reviewer's concern, we therefore added further explanations in the figure caption (lines 818-820).

For Figure 1d: we agree with the reviewer that from this data, nitroxoline seems bacteriostatic against intracellular *Salmonella* Typhi. CFUs drop only by 11.0% for *S. Typhi* 5878 and 9.2% for *S. Typhi* 2601 respectively, when cells are compared before and after treatment (whereas CFU drop is 33.8% and 34.4% respectively, when treated cells are compared to solvent control). We hence described in the Results the significant intracellular effect of nitroxoline compared to the solvent control, without commenting on its bacteriostatic or cidal activity against this organism. We would also like to point out that bacteriostatic antibiotics at therapeutic concentrations can also stall proliferation and be effective against intracellular bacteria, potentially in synergy with the host immune system (PMID: 31060222).

Figure 1c: we deleted the figure key since the number of strains is already indicated for each MIC and bacterial family. We determined MIC₅₀ and MIC₉₀ values for each species (all available as Source Data of Fig. 1c), and indicated MIC₅₀ values in the revised version of Fig. 1c (see also point 6).

16. Figure 2 & ext Fig.3: Figure 2a: FICI would be preferable since this is the standard in the field of combinations. Figure S3: which organism was used in this experiment? 'One of the most potent synergies of nitroxoline was with the OM-targeting drug colistin', requires validation, colistin does not look synergistic, FICI should be calculated. 2-fold is within the standard error range of these experiments. The drug concentration range should have been adjusted since many antibiotics did not inhibit growth alone, and therefore interactions could have been missed. The grey scale is difficult to assess. There seems to be synergy with piperacillin, rather than antagonism like the other beta-lactams? Antagonism is noteworthy, and worth highlighting to the clinical field, yet this is not well emphasized in the manuscript. Figure 2 c, statistics are not shown. Suggest removing 'animal'. Panel b is confusing. The authors could include a single strain as an example and move the others to the supplementary. Why was this measured at 10.75 h growth?

As we explained above in point 7, FICI can only be calculated for checkerboard assays where the MIC is tested, i.e. a full dose-response is mapped. In our checkerboards, we deliberately chose sub-MIC concentrations for nitroxoline and many of the other drugs, to obtain a wide dose range where we could detect the most powerful synergies, effective even at low nitroxoline dosages. This is also the reason why we used the Bliss interaction score, which can be calculated when testing sub-MIC concentrations, or for drugs that do not have activity of their own but can synergize with other compounds.

We feel that repeating the combination screen including drug MICs in order to be able to calculate FICI would not significantly improve the manuscript's message. This holds particular true for the synergy with colistin, which we already validated *in vitro* and *in vivo* against colistin-resistant isolates (Fig. 2b+c). However, to address the reviewer's point, we repeated the colistin-nitroxoline checkerboard adjusting the doses to map the dose-response from no to full growth inhibition, and calculated both Bliss and FICI interaction scores, confirming the synergy (see graph).

For piperacillin, the interaction with nitroxoline seems bimodal, synergistic at high and antagonistic at low nitroxoline concentrations. This is a possible scenario in drug-drug interactions, which translates in a neutral cumulative interaction score, but is evident when inspecting the checkerboard. This is why we included all checkerboards (ED Fig. 3) and their replicates (Supplementary Fig. 1). This is an interesting result, but we feel that more speculation on this specific interaction would fall beyond the manuscript scope. We do however agree with the reviewer's point on the clinical relevance of antagonisms, which we now mention in the discussion (lines 364-366).

Regarding Figure 2b: For validation of the colistin-nitroxoline synergy, we show resensitization of colistin-resistant strains with different and clinically relevant colistin-resistance mechanisms (*mcr-1* expression; *pmrAB* or *mgrB* mutations). We hence think that no data from Fig. 2b should be removed. The 10.75 h time point was selected as the transition between exponential and stationary phase. Furthermore, we also confirmed colistin-nitroxoline synergy across different species employing disk diffusion in *A. baumannii*, *E. coli* strain and *K. pneumoniae* strains (Supplementary File).

Regarding Fig. 2c: statistics are presented in caption (line 859-861).

Action points: we now motivate our selection of drug concentrations and of the Bliss interaction score in the Methods (lines 515-519). We provide here a new checkerboard of the colistin-nitroxoline interaction, mapping the dose-response for both drugs and calculating Bliss and FICI scores (see graph above). We further confirmed the synergy across species (*A. baumannii*, *E. coli*, *K. pneumoniae*) and in disk diffusion. These new results are shown in Supplemental Figure 3.

We now indicate in the caption of ED Fig. 3 that these experiments (related to Fig. 2a) were conducted in *E. coli* BW25113 (line 944) and increase the overall darkness of the grey scale to improve readability. We rephrased the Discussion to highlight the clinical relevance of the antagonisms with beta-lactams (lines 359-362). We rephrase Fig. 2b caption to indicate that the endpoint chosen corresponded to the transition between exponential and stationary phase (lines 845). We removed “animal” from Fig. 2c caption as suggested.

17. Figure 3: Panel a is poorly descriptive. Is this looking for changes in abundance or T_m differences? Nature of replicates etc not included.

We modified axes and scaling for Fig. 3a. Numbers and nature of replicates are now mentioned in the figure legend (lines 864-865, 872). Please see also answers to reviewer’s point 2, 9 and 13 for more detail.

18. Method detail could be improved throughout, avoiding ref to previous studies, which makes the manuscript very difficult to review. E.g., Line 487: Method detail is lacking, nature of replicates is essential. It is unclear how the samples were measured.

Action points: we added information in the Methods section for the combinatorial screen (lines 515-519) and protein digestion, peptide labelling, MS-based proteomics and data analysis for 2D-TPP (lines 541-566 for). All codes necessary for data analysis are stored in publicly accessible databases. We now specify in the methods that, as usually for 2D-TPP, the five conditions (four drug concentrations and vehicle control) are treated as distinct experiments for each of the 10 temperatures (lines 864-865). This ensures feasibility (i.e. number of samples for MS-based proteomics) and result interpretation assuming a sigmoidal dose-response relationship for protein abundance for each temperature.

19. Figure 5 – panel e, the scale is not appropriate, and it is difficult to interpret the outcomes of the experiment.

We rescaled the graphs in Fig. 5e to avoid data points at the edges of the graph to improve results readability for resistant and sensitive strains.

20. Line 62 – in the clinic?

Yes. We added in “patient isolates” (line 67).

21. Line 251 – What is the frequency of resistance?

FOR was calculated (Results, lines 273-278 and Supplementary Table 5). Please refer also to point 11 for more details.

22. Line 538: bacteria were grown ON and harvested at 0.2?

Bacteria were grown overnight in MH II medium, re-inoculated in fresh medium on the next day and harvested at an optical density OD_{600nm} of 0.2. We clarified this in the Methods section accordingly (lines 603-605).

23. Line 605 – move this section after the proteomics.

We are not sure if we understand the rationale of this suggestion: chemical genetics on *E. coli* Keio collection - described at lines 664-678 - is shown in Fig. 3, whereas proteomics of resistant mutants (that is not related to chemical genetics) is shown in Fig. 5. In the methods, we tried to reflect the progression of the data shown in the main text/figures, therefore we would like to preserve this order. We remain of course open to any suggestion/clarification.

Reviewer #2:

The authors of the manuscript “Uncovering nitroxoline activity spectrum, mode of action and resistance across Gram-negative bacteria” have conducted an impressive study. This manuscript examines the broad-spectrum applicability of the quinolone antibiotic, nitroxoline. The authors evaluated the *in vitro* efficacy of nitroxoline against a diverse and large panel of pathogens, including multidrug-resistant (MDR) clinical bacterial strains. They demonstrated that nitroxoline possesses species-specific bactericidal activity and utilized systems-based approaches to elucidate its antibacterial mode of action. Additionally, the study explores the development of resistance to nitroxoline, primarily through efflux mechanisms, across various Gram-negative bacterial species.

However, in the context of existing literature, there appears to be considerable overlap with previously established knowledge on nitroxoline. Although the authors have conducted commendable work, particularly considering the sample size and the application of systems-biology approaches, the novelty of the study is limited.

Firstly, the antibacterial activity spectrum of nitroxoline has already been reported against various bacterial pathogens beyond *E. coli*, such as *K. pneumoniae*, *P. aeruginosa*, *A. baumannii*, *S. aureus*, and *Enterococci* sp. (References: PMID: 31423123, 32899218, 32364820, 31423123, 31292653). The species-specific bactericidal nature of nitroxoline has also been demonstrated against other bacteria (Ref: PMC6395908). Additionally, the metal chelation activity of nitroxoline against zinc and copper has been previously documented (References: PMID: 31423123, 22926564). The ability of nitroxoline to disrupt the outer membrane has also been shown (Ref: PMID: 38085067). Moreover, the role of RND efflux pumps in resistance development against nitroxoline has been shown in literature as has also been discussed in the manuscript (PMID: 38063385; 31633764).

We acknowledge the opinion of the reviewer and the remark on existing literature on nitroxoline. We are well aware of all the mentioned studies and had cited and/or discussed most of them. We address the reviewer’s point on novelty by adding direct citations to these papers when discussing specific results in the manuscript (activity spectrum, cidal, OM and metal perturbation, RND pumps), highlighting the advancement that our study provides on each point as outlined below. Overall, though these papers cover similar topics, they either have a different focus, are far from our scale in terms of species tested and mechanistic exploration, or remain largely descriptive, lacking direct evidence of the points made (e.g. metal perturbation).

Regarding the references on nitroxoline’s antibacterial activity (PMID: 31423123, 32899218, 32364820 and 31292653), we had either cited them directly (PMID: 32364820 as ref. 25; PMID: 31292653 as ref. 67) or indirectly by citing an extensive 2022 review detailing the assay used and number of strains tested for each study (PMID: 35907036 as ref. 18). Our study is superior to all these instances by number of species and strains tested as well as number of orthogonal assays used for MIC determination, as outlined below.

Bactericidal activity (PMC6395908) was tested against *Mycoplasma* spp. and *Ureaplasma* spp. isolates, which have no cell wall and hence cannot be compared to our data, especially when considering the effect of nitroxoline on the OM. However, we agree that species-specificity of cidal/static activity should be assessed with a larger number of strains per species. Therefore, to address this and point 5 by reviewer 1, we toned down this aspect of our findings in the abstract and discussion, where we now cite PMC6395908.

PMIDs 31423123 and 22926564 (ref. 23) analyze the indirect effect of cation titration on bacterial phenotypes such as MIC and biofilm formation, whereas we described proteome-wide effects of nitroxoline on metal homeostasis and directly determined the intracellular metal

concentration upon nitroxoline exposure. Based on our XRF and 2D-TPP data, we were able to show that nitroxoline not only acts as a chelator outside the bacterial cell, but also leads to intracellular intoxication by copper and zinc in *E. coli*, which was previously unknown. Our data therefore represents an advancement compared to previous knowledge in: (i) resolution and breadth of phenotypic effects (2D-TPP); (ii) strength of the evidence of metal intoxication via direct measurement of intracellular metals; (iii) novelty of the finding of copper and zinc intracellular increase.

He et al. (PMID: 38085067, ref. 48), which focused on the interaction of nitroxoline's action on NDM-1-positive bacteria, showed the interaction of nitroxoline with purified LPS *in vitro* and disruption of the OM with the same NPN assay that we use (Fig. 3c), on another *E. coli* strain. Unfortunately, we were not able to directly compare our results for this assay, since raw data were not provided with the paper, we could not identify a mean or median fluorescence value for each condition, and the nature of their only control was not specified (we used chloramphenicol, polymyxin B - with or without NPN - and NPN-only, all with and without cells). Overall, while He et al. provide valuable insights on the interplay between nitroxoline, calcium and LPS, we show nitroxoline genome- and proteome-wide effects on LPS- and OM biosynthesis, thereby providing complementary and novel results. Specifically, we found that mutants in LPS biosynthesis and transport were more sensitive to nitroxoline, whereas mutants involved in the first reactions of heptose incorporation into LPS inner core biosynthesis (*gmhA*, *gmhB* and *waaC*) were more resistant (Fig. 3b). This suggests the heptosyl-Kdo₂ moiety of the LPS inner core as a minimum requirement for nitroxoline activity, since deletion of mutants catalyzing downstream LPS biosynthetic reactions, starting with *waaF*, were more sensitive to nitroxoline.

We had acknowledged and cited both references on the role of RND efflux pumps in nitroxoline resistance (PMID: 38063385 as ref. 30; 31633764 as ref. 29). However, our assessment of resistance mechanisms is much more comprehensive in terms of species tested, but also within-species for diversity of ancestor and evolved resistant strains tested. For example, ref. 30 includes a large WGS dataset on 13 resistant *E. coli* and 17 *K. pneumoniae* strains, as well as proteomics on three *E. coli* resistant strains, all evolved from one parental strain. Ref. 29 provides a valuable WGS dataset on 12 *E. coli* resistant strains. We provide WGS data on 12 *E. coli* and eight *K. pneumoniae* resistant strains (Fig. 5a), evolved from four and seven, respectively, different sensitive ancestors, as well as six *A. baumannii* resistant strains from five distinct ancestors. We performed proteomics on two *A. baumannii*, three *E. coli* and seven *K. pneumoniae* and on their sensitive ancestors (eight in total). It is true (and in the opinion of ourselves and reviewer 1, positive) that we confirm RND efflux pumps and porins as major alterations. However, in contrast to other studies we were able to (i) generalize resistance mechanisms to the species level thanks to the diversity of sensitive ancestors considered; (ii) assess a much larger range of mutations; (iii) evaluate the cross-species conservation of resistance mechanisms, particularly including *A. baumannii* against which we propose nitroxoline as new resource.

Furthermore, we were able to assess the clinical relevance of these mechanisms by (i) systematically comparing the mutation landscape in RND pump regulators in experimentally evolved and clinical nitroxoline-resistant isolates and (ii) by presenting the first case of nitroxoline resistance evolution in *K. pneumoniae* in a patient. This allowed us to demonstrate the *in vivo* relevance of *oqxR* mutations and identify G60_67dup as the most common mutation in both experimentally evolved and clinical isolates.

Action points: we highlight open questions as well as our novel findings and advancements in the abstract (lines 33-38), introduction (lines 52-65 and 75-86) and discussion (lines 364-382; 391-403, also responding to suggestions of reviewer 1). We added all references mentioned

by the reviewer that we had not already cited (except PMID: 31423123, which is already cited in the comprehensive review of nitroxoline's antibacterial spectrum as ref 18). Specifically, we now cite PMC6395908 in the discussion, where we expand and tone down the species-specificity aspect of nitroxoline cidality (lines 391-403).

Reviewer #3:

In this manuscript the authors interrogate the mechanism of action of the antibiotic nitroxoline. The authors set out to test if nitroxoline could be repurposed against a wide range of bacterial pathogens to expand its therapeutic profile. These results revealed a promising antimicrobial profile against Enterobacteriales, Acinetobacter, and intracellular Salmonella. The authors also report the surprising finding that nitroxoline is bactericidal in certain cases and that it potentiates the therapeutic benefit of colistin in a Galleria model of Klebsiella infection. Using a chemical genetics approach, the authors suggest that nitroxoline affects LPS biosynthesis. In addition, nitroxoline exposure led to an increase in the abundance of metal transporters and regulators which are involved in metal homeostasis, suggesting a role for nitroxoline in metal intoxication. The authors conclude the manuscript with identification of mutations that provide nitroxoline resistance in multiple bacterial species. The authors conclude that these findings suggest that nitroxoline does not have a protein target and that its therapeutic window is now wider for the treatment of infectious diseases. The manuscript is well written, the experiments are well controlled, and the results are provocative and will be of interest to a wide audience. I have only one question for the authors to consider.

1. It is unusual that metal import and efflux mutants are both more sensitive to nitroxoline. Can the authors provide text to explain this?

We thank the reviewer for the positive comments! From our chemical genetic data, we can observe a loss-of-fitness for mutants encoding for iron import. This is compatible with the upregulation of iron importers and uptake mechanisms in 2D-TPP, which we interpret as a response to Fe-S damage, associated with zinc and copper intoxication (ref. 48-51).

For zinc and copper, both metal exporters (e.g. ZntA and CopA) are increased/stabilized in 2D-TPP, consistently with a defense response to the zinc and copper overload which we directly measured via nano-XRF. However, in chemical genetics we find a loss of fitness only for the mutants of the regulators ($\Delta cueR$, $\Delta zntR$). The reason for this is unclear; but we would hypothesize, that, if the main exporter is deleted but the regulator is preserved, it can still trigger/repress alternative efflux mechanisms.

Regarding zinc importers, ZnuA and ZnuC are decreased/destabilized in 2D-TPP, which we interpret as a response of Zur stabilization upon zinc excess. However, we could not detect any significant fitness change in Znu subunit mutants, except for a mild loss of fitness of $\Delta znuB$. This might suggest that deletion of one subunit is not sufficient to substantially limit zinc import. The chemical genetic results suggest that zinc efflux and other drug-response mechanisms, e.g. active drug efflux or iron uptake to regenerate Fe-S clusters, are more important than limitation of zinc uptake upon nitroxoline exposure.

The reviewer's comment made us realize that we did not highlight $\Delta mntH$ among the mutants losing fitness with nitroxoline. MntH is a manganese importer that we find increased in 2D-TPP as a known response to iron deprivation and Fe-S damage, where manganese can be used as iron replacement. Accordingly, we find $\Delta mntH$ more sensitive to nitroxoline. We now show this in Fig. 3b.

Action points: we summarize the outlined points on the role of metal export/import (particularly for zinc) in the Results (lines 234-245, 255-256) and now highlight $\Delta mntH$ in Fig. 3b.